# Dynamic Chunking for End-to-End Hierarchical Sequence Modeling

**Sukjun Hwang[1], Brandon Wang[2], Albert Gu[1,2]**
[1]Carnegie Mellon University, [2]Cartesia AI

## Abstract

Major progress on language models (LMs) in recent years has largely resulted from moving away from specialized models designed for specific tasks, to general models based on powerful architectures (e.g. the Transformer) that learn everything from raw data. Despite this trend, pre-processing steps such as tokenization remain a barrier to true end-to-end foundation models. We introduce a collection of new techniques that enable a **dynamic chunking** mechanism which automatically learns content- and context- dependent segmentation strategies learned jointly with the rest of the model. Incorporating this into an explicit **hierarchical network (H-Net)** allows replacing the (implicitly hierarchical) tokenization–LM–detokenization pipeline with a single model learned fully end-to-end. When compute- and data- matched, an H-Net with one stage of hierarchy operating at the byte level outperforms a strong Transformer language model operating over BPE tokens. Iterating the hierarchy to multiple stages further increases its performance by modeling multiple levels of abstraction, demonstrating significantly better scaling with data and matching the token-based Transformer of twice its size. H-Nets pretrained on English show significantly increased character-level robustness, and qualitatively learn meaningful data-dependent chunking strategies without any heuristics or explicit supervision. Finally, the H-Net's improvement over tokenized pipelines is further increased in languages and modalities with weaker tokenization heuristics, such as Chinese and code, or DNA sequences (nearly $4\times$ improvement in data efficiency over baselines), showing the potential of true end-to-end models that learn and scale better from unprocessed data.

## 1 Introduction

A broad goal of deep learning is to learn meaningful patterns from raw data, automatically extracting features and building abstractions in an end-to-end fashion. However, fixed-vocabulary **tokenization**, the process of compressing raw text into predefined chunks through algorithms such as byte-pair encoding (BPE) (Sennrich et al., 2015; Kudo & Richardson, 2018), remains a pervasive handcrafted preprocessing step in modern language models (LMs) (Grattafiori et al., 2024; Brown et al., 2020). Tokenization comes with a host of well-documented drawbacks, from poor character-level understanding to lack of meaning and interpretability to degraded performance on complex languages and modalities (Petrov et al., 2023; Ahia et al., 2023; Belinkov & Bisk, 2017; Sun et al., 2020; Clark et al., 2022). Replacing the tokenization–LM–detokenization pipeline with a single end-to-end model would also adhere better to the spirit of deep learning, ideally scaling more powerfully with data and parameters (c.f. *the bitter lesson*) (Sutton, 2019; Perić, 2025). However, tokenization remains an indispensable component of language models and other sequential data for its ability to compress and shorten sequences; as of yet, no *end-to-end* tokenizer-free model has matched the performance of tokenizer-based language models when matched for computational budget.

A line of recent works has turned to overcoming tokenization in autoregressive sequence models, which requires addressing a series of difficult technical challenges:[1]

- Direct byte-level language modeling with isotropic architectures[2] can be improved with efficient sequence models such as MambaByte (Wang et al., 2024), but still incur prohibitive computational costs while underperforming tokenized models in compute-matched settings.

---

[1]An extended related work can be found in Section B, which is summarized in Table 4.

[2]Non-hierarchical models with repeated blocks, such as the standard Transformer (Vaswani et al., 2017).

- To improve efficiency, hierarchical architectures such as Hourglass Transformer (Nawrot et al., 2022) and MegaByte (Yu et al., 2023) use small byte-level models to compress raw inputs into subsampled sequences, which are then processed with a more powerful standard language model. However, simple pooling strategies such as compressing every $k$ inputs are not data-dependent, and perform poorly on modalities with variable information rates such as language.

- SpaceByte (Slagle, 2024) and Byte Latent Transformer (Pagnoni et al., 2024) introduce data-dependent chunking strategies such as delimiter- or entropy-based heuristics. These heuristics, however, rely on auxiliary *external* boundary predictors, and are therefore modality-specific and not fully end-to-end.

- Although jointly trainable boundary predictors are the ideal solution, they require optimizing discrete selection operations without supervision, which is fundamentally a challenging problem. Consequently, existing end-to-end approaches (Nawrot et al., 2023) exhibit training instabilities that preclude scaling beyond small models or nesting multi-level hierarchies.

Fundamentally, creating a tokenizer-free architecture requires incorporating the data chunking process directly into the model, while overcoming challenges in efficiency, learnability, and stability at scale.

DYNAMIC CHUNKING: END-TO-END SEQUENCE MODELING WITHOUT TOKENIZATION

In this work, we introduce an end-to-end **hierarchical network (H-Net)** that compresses raw data through a recursive, data-dependent **dynamic chunking (DC)** process (Figure 1). H-Nets match the efficiency of tokenized pipelines while substantially improving modeling ability, by replacing handcrafted heuristics with content-aware and context-dependent segmentation learned from data.

**Hierarchical Processing.**   The H-Net adopts the hierarchical architecture from prior work (Goel et al., 2022; Nawrot et al., 2022; Slagle, 2024), resembling an autoregressive U-Net (Ronneberger et al., 2015): (i) raw data is processed by a small **encoder network**, (ii) then downsampled and passed through a **main network** operating on compressed chunks, (iii) and finally upsampled before being passed through a **decoder network** operating on the original resolution. This modularity creates a natural processing hierarchy where outer stages capture fine-grained patterns while inner stages operate on coarse representations akin to traditional tokens. Crucially, while the main network contains the bulk of parameters and can be any standard architecture designed for operating on tokenized language—such as a Transformer (Vaswani et al., 2017) or state space model (SSM) (Gu & Dao, 2024)—we show that the encoder and decoder networks are strongly improved by using SSMs, which have an inductive bias for compression (Gu, 2025).

**Dynamic Chunking.**   H-Net's core is a novel dynamic chunking (DC) mechanism which interfaces between the main network and the encoder/decoder networks, learning how to segment data while using standard differentiable optimization. DC is composed of two complementary new techniques: (i) a **routing module** which predicts boundaries between adjacent elements through a similarity score (ii) and a **smoothing module** which interpolates representations using the router's outputs, attenuating the effect of uncertain boundaries and significantly improving learnability. By combining these with a new auxiliary loss function that targets desired downsampling ratios, and modern techniques for gradient-based learning of discrete choices (Fedus et al., 2022; Bengio et al., 2013), DC lets an H-Net learn how to compress data in a fully end-to-end fashion.

**Signal Propagation.**   We introduce several architectural and training techniques to improve stability and scalability during end-to-end optimization. These include: (i) carefully placing projections and normalization layers to balance signal propagation between interacting sub-networks, and (ii) adjusting optimization parameters for each layer based on its dimensionality and effective batch size, which changes between stages of the hierarchical structure.

Altogether, H-Net learns segmentation strategies *optimized jointly* with the main backbone, dynamically compressing input vectors based on contextual information into meaningful chunks. H-Net represents the first truly end-to-end, tokenizer-free language model: with a single stage of dynamic chunking, a *byte-level H-Net* **matches the perplexity and downstream performance of a strong BPE-tokenized Transformer** at sizes exceeding 1B parameters. Empirically, the dynamic chunking module naturally compresses data to a similar resolution as BPE tokenizers (4.5-5 bytes/chunk) and qualitatively learns meaningful boundaries, all without any external supervision or heuristics.

HIERARCHICAL CHUNKING: FROM DATA TO ABSTRACTIONS

Beyond addressing tokenization, H-Net improves general sequence modeling across a wide range of settings. Subword tokenization in language models is a special case of *chunking*—the process of building higher-level abstractions from low-level data—and is a central component of intelligence.[3] Crucially, because H-Net is fully end-to-end, **it can be iterated recursively: the main network can itself be an H-Net**. Intuitively, more stages of chunking represent higher order meanings; just as characters can be combined into words, words can be combined into clauses, sentences, and beyond. Iterating the hierarchy should therefore lead to even more efficient use of compute and parameters, and more effective reasoning over compressed representations.

Recursive H-Nets represent a new class of foundation model architectures that not only overcome tokenization, but discover and operate over abstractions learned from raw data, leading to higher-quality models with less pre-processing.

Iterating the 1-stage H-Net to 2 hierarchical stages further improves its capabilities and strongly outperforms all baselines, with steeper training curves and better scaling with data. A byte-level 2-stage H-Net overtakes the perplexity of a strong tokenized Transformer after just 30B training bytes, with the gap widening throughout training, and matches the downstream evaluations of the tokenized Transformer of twice its size.

Finally, H-Nets realize the benefits of overcoming tokenization:

- *Robustness*. Without special data mixes, the pretrained H-Net is dramatically more robust to textual perturbations than the token-based Transformer, as evaluated on the noisy HellaSwag suite of benchmarks.

- *Interpretability*. Qualitative visualizations of learned boundaries reveal that H-Net automatically discovers semantically coherent units without explicit supervision, validating that end-to-end learning successfully detects the structural patterns traditionally imposed through handcrafted tokenization.

- *Other languages*. H-Net's improvements are even more pronounced on languages without obvious segmentation cues, including Chinese and code ($59.9 \rightarrow 66.3$ on XWinograd-zh compared to tokenized Transformer) and DNA language modeling ($3.6\times$ improved data efficiency compared to isotropic models).

## 2 H-NET ARCHITECTURE

H-Nets are hierarchical U-Net-like networks, but with data-dependent *dynamic subsampling* that is learned end-to-end together with the rest of the model. We first introduce H-Net's hierarchical architecture for multi-level processing (Section 2.1), then present our dynamic chunking mechanism that learns content-aware compression through standard optimization (Section 2.2). Additional details appear in the appendix (Section C), including: (i) architectural design principles; (ii) more explanations about the dynamic chunking mechanism; (iii) optimization and architectural enhancements for hierarchical sequence modeling; and (iv) preservation of autoregressive properties.

### 2.1 ARCHITECTURAL OVERVIEW

H-Net employs a hierarchical architecture comprising three primary components – encoder networks ($\mathcal{E}$), main network ($\mathcal{M}$), and decoder networks ($\mathcal{D}$) – where each component is implemented with a stack of sequence mixing layers (*e.g.,* Transformers or state space models). In its simplest form, a single-stage H-Net consists of one encoder network, one main network, and one decoder network. Crucially, the architecture's key characteristic lies in the main network's unique property: the main network can be another complete H-Net, enabling recursive construction of multi-level hierarchies.

This recursive design allows H-Net to scale to arbitrary depths. In an $S$-stage model, we denote components at each stage using superscripts: encoder networks as $\mathcal{E}^s$ and decoder networks as $\mathcal{D}^s$ for stages $0 \leq s < S$, with the main network $\mathcal{M}$ residing only at the final stage $s = S$. For example, a two-stage model contains $\mathcal{E}^0$, $\mathcal{E}^1$, $\mathcal{M}$, $\mathcal{D}^1$, and $\mathcal{D}^0$, as illustrated in Figure 1-(Left). Throughout this paper, we use superscripts to denote stage indices, though we omit them when all variables within an equation belong to the same stage.

---

[3]Chunking is a formal concept from cognitive psychology central to human memory and cognition, and is the inspiration for this work's terminology.

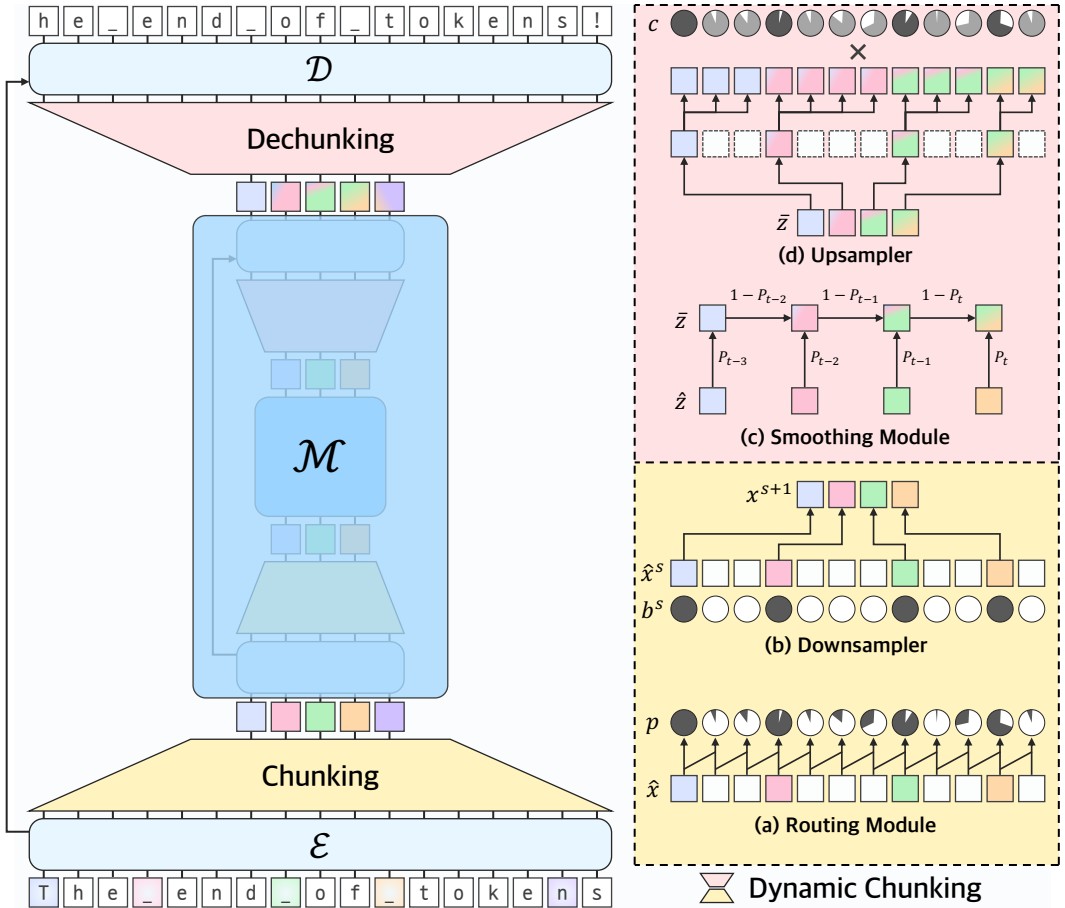

Figure 1: **(left)** Architectural overview of H-Net with a two-stage hierarchical design ($S = 2$). **(right)** Dynamic Chunking (DC). **(bottom-right)** Key components of a chunking layer: (a) a routing module for dynamically drawing chunk boundaries, and (b) a downsampler that selectively retains vectors based on boundary indicators, reducing sequence length while preserving semantically significant positions. **(top-right)** Key components of a dechunking layer: (c) a smoothing module for converting discrete chunks into interpolated representations, and (d) an upsampler that restores compressed vectors to their original resolution based on boundary indicators. Linear in equation equation 3 and STE in equation equation 9 are omitted in the illustration for brevity.

Drawing inspiration from the U-Net architecture (Ronneberger et al., 2015), H-Net progressively compresses input sequences into fewer vectors with richer semantic embeddings through a chunking layer, processes these representations in the main network, then decompresses the sequence back to its original resolution using a dechunking layer. Unlike traditional U-Net designs, however, H-Net dynamically determines chunking boundaries rather than using fixed-size pooling operations. The overall pipeline can be formalized as:

$$\hat{x}^s = \mathcal{E}^s(x^s), \qquad \hat{z}^S = \mathcal{M}(x^S), \qquad \hat{z}^s = \mathcal{D}^s(z^s), \qquad (1)$$

where the chunking layer and the dechunking layer operations are defined as:

$$(x^{s+1}, p^s) = \mathsf{Chunk}(\hat{x}^s), \qquad (2) \qquad z^s = \mathsf{Dechunk}(\hat{z}^{s+1}, p^s) + \mathsf{Linear}(\hat{x}^s). \qquad (3)$$

The initial input to the model is $x^0 \in \mathbb{R}^{L^0 \times D^0}$ where $L^0$ is the input sequence length and $D^0$ is the embedding dimension. Intuitively, $p^s \in [0, 1]^{L^s}$ represents the chunking router's confidence that the token should be passed into the main stage.[4] This value is essential for both the chunk (Section 2.2.1) and dechunk operations (Section 2.2.2).

We further provide full details about H-Net's **design principles** in Section C.1.

---

[4]We also sometimes refer to it as a *probability*—it is interpreted as such in Section E.2—although we do not use it as a formal probability.

## 2.2 DYNAMIC CHUNKING (DC)

H-Net learns chunking boundaries through end-to-end training, allowing it to identify semantically meaningful units adaptively. Furthermore, this dynamic approach enables the model to allocate computational resources efficiently by compressing low-information regions while preserving high-information content at appropriate granularity.

### 2.2.1 CHUNKING LAYER

The chunking layer (Chunk in equation equation 2) contains a routing module and downsampler, as illustrated in Figure 1-(bottom-right).

**Routing Module.** In natural data, meaningful boundaries tend to emerge at points of contextual or semantic shift. From this observation, we add an inductive bias by measuring the similarity between adjacent representations: when context changes, consecutive vectors should exhibit lower similarity. The routing module implements this intuition through cosine similarity between adjacent encoder outputs $\hat{X}$, it calculates boundary probabilities $p_t$ and boundary indicators $b_t$ as follows:

$$q_t = W_q \hat{x}_t, \quad k_t = W_k \hat{x}_t, \qquad p_t = \frac{1}{2}\left(1 - \frac{q_t^\top k_{t-1}}{\|q_t\|\,\|k_{t-1}\|}\right) \in [0,1], \quad b_t = \mathbb{1}_{\{p_t \geq 0.5\}}, \qquad (4)$$

where $p_1 = 1.0$ by definition, ensuring the sequence begins with a boundary. This formulation scales cosine similarity into a boundary score or probability: ideally, when consecutive vectors $\hat{x}_{t-1}$ and $\hat{x}_t$ span a semantic boundary (*e.g.,* between morphemes, words, or phrases), their projections $q_t$ and $k_{t-1}$ diverge in the latent space, yielding low cosine similarity and consequently high boundary probability $p_t$.

**Downsampler.** The downsampler compresses encoder outputs $\hat{x}^s$ into a reduced set of vectors $x^{s+1}$ using boundary indicators $\{b_t^s\}_{t=1}^{L^s}$. Among potential compression strategies – including mean pooling, max pooling, or cross-attention – we adopt direct selection of boundary-marked vectors for its simplicity and effectiveness (see Section E.3.6 for ablations).

As illustrated in Figure 1-(b), this approach follows a straightforward selection rule: vectors where $b_t = 1$ are retained in the compressed sequence $x^{s+1}$, while those where $b_t = 0$ are discarded. Likewise, the same downsampler applies to boundary probabilities, compressing $p^s$ into $P^{s+1}$ for use in a dechunking layer (see Section 2.2.2).

### 2.2.2 DECHUNKING LAYER

The dechunking layer (Dechunk in equation equation 3) consists of a smoothing module and up-sampler, as illustrated in Figure 1-(top-right).

**Smoothing Module.** The critical challenge in training a dynamic chunking module lies in the discrete nature of chunk boundaries, which impedes gradient flow during backpropagation. We introduce the smoothing module as a technique to address this problem. As illustrated in Figure 1-(c), this component transforms discrete chunking operations into differentiable computations by creating smooth interpolations between chunks. Concretely, the smoothing module applies an exponential moving average (EMA) with the following definition:

$$\bar{z}_t = P_t \hat{z}_t + (1 - P_t)\bar{z}_{t-1}. \tag{5}$$

In Section C.2.1, we describe several roles of the smoothing module and provide Figure 4 with more explanations.

**Upsampler.** We carefully design the upsampler (see Figure 1-(d)) that decompresses $\bar{z}^{s+1}$ to match the original resolution of inputs in the previous stage $z^s$ with the following definition:

$$c_t = p_t^{b_t}(1 - p_t)^{1-b_t} = \begin{cases} p_t & \text{if } b_t = 1, \\ 1 - p_t & \text{otherwise,} \end{cases} \tag{6} \qquad \tilde{z}_t = \bar{z}_{\sum_{k=1}^t b_k}, \tag{8}$$

$$\mathsf{STE}(c_t) = c_t + \text{stopgradient}(1 - c_t), \tag{7} \qquad \mathsf{Upsampler}(\bar{z}, c)_t = \mathsf{STE}(c_t) \cdot \tilde{z}_t. \tag{9}$$

Each component serves a specific purpose in enabling stable end-to-end learning, which is described in Section C.2.2.

### 2.2.3 RATIO LOSS

Without explicit regularization, the model may converge to trivial solutions: either retaining nearly all vectors (negating computational benefits) or compressing excessively (losing critical information). Inspired by load balancing mechanisms in Mixture-of-Experts (MoE) models (Fedus et al., 2022), which face similar challenges in maintaining balanced expert utilization, we introduce a ratio loss to guide compression:

$$\mathcal{L}_{\text{ratio}} = \frac{N}{N-1}\left((N-1)FG + (1-F)(1-G)\right), \qquad F = \frac{1}{L}\sum_{t=1}^{L} b_t, \quad G = \frac{1}{L}\sum_{t=1}^{L} p_t, \quad (10)$$

where $F$ represents the fraction of vectors actually selected, $G$ denotes the average boundary probability, and $N$ controls the target compression ratio. Mechanistically, although $F$ is not differentiable, the network can be trained toward targeted compression ratios through $G$, which provides continuous feedback. When $F = G$, the loss attains a minimum of $\mathcal{L}_{\text{ratio}} = 1$ when $F = G = \frac{1}{N}$. Interestingly, the loss can theoretically fall below 1 when $F \neq G$ (*e.g.*, $F = \frac{1}{N} + \epsilon$ and $G = \frac{1}{N} - \epsilon$), which we indeed observe during training. Despite this theoretical possibility, the loss effectively guides the model toward the desired compression ratio in practice. In practice, as our architectural design encourages the routing module to make confident decisions (*i.e.,* boundary probabilities approaching 0 or 1), $F$ naturally converges toward $G$, and the loss effectively guides the model toward the desired compression ratio. Notationally, we sometimes use $(N^0, N^1, \ldots, N^{s-1})$-DC to denote the full dynamic chunking mechanism together with its targeted chunking ratios.

Combined together with the autoregressive prediction loss (*i.e.,* $\mathcal{L} = \mathcal{L}_{\text{AR}} + \alpha \sum_{s=0}^{S-1} \mathcal{L}_{\text{ratio}}^s$), this mechanism preserves content-adaptive compression: the model learns which vectors to retain based on semantic importance rather than following predetermined patterns, distinguishing H-Net from fixed compression schemes. We fixed $\alpha = 0.03$ in all experiments in this paper as it provides a good balance between prediction accuracy and chunking efficiency; however, in other settings, it may be important to choose this hyperparameter more carefully.

## 3 EXPERIMENTS

We first describe our general experimental protocol for language modeling, used for the majority of our experiments. In Section 3.1, we evaluate on a high-quality English dataset, showing significantly stronger performance than baselines, as well as improved robustness and interpretability from avoiding tokenization. In Section 3.2, we extend our evaluation to diverse datasets including Chinese, and code, with even larger performance improvements, demonstrating H-Net's versatility as a general sequence model architecture. In the appendix, we share further details about the experiments in this section (see Section D), and provide more experiments and ablation studies (see Section E).

**Models.** We compare against a standard tokenized Transformer following the Llama architecture (Touvron et al., 2023b; Grattafiori et al., 2024).[5] We additionally compare against several byte-level baselines:

- **MambaByte** (Wang et al., 2024) and **LlamaByte** are isotropic models using pure Mamba-2 layers and pure Transformer layers, respectively.

- **SpaceByte** (Slagle, 2024) represents the canonical hierarchical architecture with external boundary supervision, which chunks on spaces and "space-like" bytes.[6] On English, the space-like delimiter heuristic empirically has an average ratio of 6.0 bytes per chunk.

- **SpaceByte++** is our modification of SpaceByte that includes our architectural modifications to the hierarchical structure (from Section 2.1). In particular, it changes the outer encoder/decoder networks to use Mamba-2, and modifies the layer counts and widths slightly to match the H-Net models below.

- **H-Net (space)** and **H-Net (pool)** differ from our full H-Net only through the chunking function. H-Net (space) further improves SpaceByte++ with our training improvements to the network (Sec-

---

[5]This was called the "Transformer++" in Gu & Dao (2024); since by now it is firmly established, we remove the "++".

[6]BLT is another architecture with external supervision using entropy instead of delimiters, but is unfortunately too complex to set up and control as a baseline. We believe that the delimiter-based method is highly competitive. See Section B.1.3.

Table 1: **Architectures for main language models, all data-/FLOP-matched.** $\mathcal{E}^0, \mathcal{D}^0, \mathcal{E}^1, \mathcal{D}^1,$ $\mathcal{M}$. T and M denote a Transformer and a Mamba-2 layer, respectively. For hierarchical byte-level models, the Tokenizer column lists the chunking mechanism. The numbers before DC indicate downsampling factor $N$ in equation equation 10; for example, (3,3)-DC denotes $N^0 = N^1 = 3$. The BPIC (Bytes-Per-Innermost-Chunk) measure shows that each chunk dynamically determined by our 1-stage comprises similar number of bytes to the GPT-2 tokenizer, despite aiming for $N^0 = 6$. All Transformer layers in $\mathcal{E}$ or $\mathcal{D}$ networks, as well as LlamaByte, use Sliding Window Attention (SWA) with a window size of $1024$. Just as in the original Mamba (Gu & Dao, 2024) and Mamba-2 (Dao & Gu, 2024) blocks, our Mamba-2 layers have roughly $6(D^s)^2$ parameters and Transformer layers have $12(D^s)^2$ parameters in stage $s$.

| Model | Input | Tokenizer | $L^0$ | BPIC ($L^S/L^0$) | #Params | Architecture | d_model (D) |
|---|---|---|---|---|---|---|---|
| **#FLOPs matched to GPT-3 *Large*** | | | | | | | |
| Transformer | Token | GPT2 | 1792 | 4.6 | 760M | T24 | 1536 |
| LlamaByte | | — | | 1.0 | 210M | T16 | 1024 |
| MambaByte | | — | | 1.0 | 190M | M28 | 1024 |
| SpaceByte | | Spacelike | | 6.0 | 570M | T8 + T16 + T8 | 768 , 1536 |
| SpaceByte++ | | Spacelike | | 6.0 | 850M | M4 + T28 + M4 | 1024 , 1536 |
| H-Net (pool) | Byte | 6-Pool | 8192 | 6.0 | 850M | M4 + T28 + M4 | 1024 , 1536 |
| H-Net (space) | | Spacelike | | 6.0 | 850M | M4 + T28 + M4 | 1024 , 1536 |
| H-Net (1-stage) | | 6-DC | | 4.8 | 680M | M4 + T22 + M4 | 1024 , 1536 |
| H-Net (2-stage) | | (3,3)-DC | | 7.0 | 870M | M4 + T1M4 + T26 + M4T1 + M4 | 1024 , 1024 , 1536 |
| **#FLOPs matched to GPT-3 *XL*** | | | | | | | |
| Transformer | Token | GPT2 | 1792 | 4.6 | 1.3B | T24 | 2048 |
| SpaceByte++ | | Spacelike | | 6.0 | 1.6B | M4 + T31 + M4 | 1024 , 2048 |
| H-Net (space) | Byte | Spacelike | 8192 | 6.0 | 1.6B | M4 + T31 + M4 | 1024 , 2048 |
| H-Net (1-stage) | | 6-DC | | 4.7 | 1.3B | M4 + T24 + M4 | 1024 , 2048 |
| H-Net (2-stage) | | (3,3)-DC | | 6.9 | 1.6B | M4 + T1M4 + T27 + M4T1 + M4 | 1024 , 1536 , 2048 |

tion C.3), in particular, adding post-network norms, residual projections, and learning rate multipliers on the outer networks. H-Net (pool) is a baseline ablating the effect of a simple chunking strategy that pools every $k$ tokens, which is expected to be weaker than all of the data-dependent chunking strategies.

- **H-Net (1-stage)** is our full H-Net method with DC learned end-to-end (Section 2.2) with compression target $N^0 = 6$.

- **H-Net (2-stage)** is our full H-Net method, iterated to two nested stages using $N^0 = 3, N^1 = 3$.

We provide results for two model scales, *Large (L)* and *XL*. Each scale is FLOP-matched to the corresponding GPT-3 (Brown et al., 2020) (*i.e.,* GPT-3 L and GPT-3 XL) variant of the tokenized Transformer (760M and 1.3B parameters respectively).

**Experimental Setup.** Following established practice (Xue et al., 2022; Wang et al., 2024; Slagle, 2024), we measure performance using bits-per-byte (BPB) to ensure comparability across different input representations. For tokenized models, this amounts to simply rescaling the total negative log likelihood of a sequence (in tokens) by the total number of bytes. In addition, we systematically control the data and compute budget for all models (see Table 1), matching all models carefully in both **bytes-per-batch** and **FLOPs-per-byte**:

- Data Budget: We train all models on the 100B token subset sampled from the FineWeb-Edu dataset (Penedo et al., 2024). All tokenizer-free models process 8192 `utf-8` encoded bytes per sequence, while the Transformer uses 1792 tokens from the GPT2 tokenizer (roughly equivalent to 8192 bytes). We use batch size 256 for all models; the total batch size is just under $0.5$M tokens per batch for the baseline BPE Transformer, roughly matching protocols from prior work (Gu & Dao, 2024).

- Compute Budget: For calculating FLOPs, we follow standard methodology (Hoffmann et al., 2022) with an extension for Mamba-2 layers (see Section D.1). We use the BPE-tokenized Transformer's #FLOPs as a reference, and the number of layers of the other models is adjusted accordingly to match the reference #FLOPs.

Training employs AdamW (Loshchilov & Hutter, 2017) optimizer with warmup-stable-decay (WSD) (Hu et al., 2024) scheduling with $10\%$ linear warmup and $20\%$ inverse-square-root de-

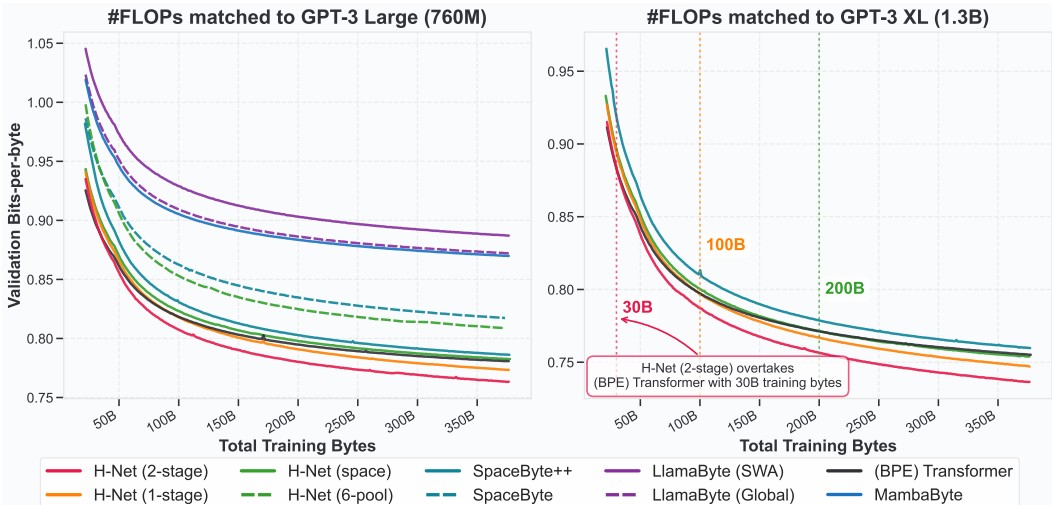

Figure 2: **Validation Bits-per-byte (BPB) throughout training** for different models at Large (760M, left) and XL (1.3B, right) scales with matched computational and data budgets for training. All models but Transformer take raw byte inputs (Transformer uses GPT-2 tokenizer). Vertical dotted lines indicate crossover points where H-Net begins to outperform Transformer with predefined BPE tokenization. From the curves we can clearly see the following: (1) all hierarchical models (*i.e.,* SpaceByte++, H-Net variants) outperform the isotropic models (*i.e.,* Transformer, MambaByte, LlamaByte); (2) dynamic chunking is more powerful than BPE tokenizers; and (3) DC is more effective than other chunking strategies. Furthermore, H-Net's 2-stage variant consistently outperforms 1-stage across both scales, demonstrating the effectiveness of deeper hierarchies. See Table 1 for architectural details.

cay (Ibrahim et al., 2024). Following Hägele et al. (2024) which recommends WSD schedulers with half the maximum learning rates as a cosine schedule, we adopt learning rates $2.5\times$ higher than GPT-3 (Radford et al., 2019) standards; this corresponds to half of the maximum learning rate used in Gu & Dao (2024), yielding $6.25 \times 10^{-4}$ for *Large*-scale models and $5.0 \times 10^{-4}$ for *XL*-scale models. Architecture details include gated MLPs (Touvron et al., 2023b) in all Transformer layers and the main network's Mamba layers, while Mamba layers in $\mathcal{E}$ and $\mathcal{D}$ are without an MLP. For Transformer layers in $\mathcal{E}$ and $\mathcal{D}$, we use Sliding Window Attention (SWA) (Beltagy et al., 2020) with the window size of $1024$. As discussed Section C.1, $\mathcal{E}$ and $\mathcal{D}$ comprise mainly Mamba-2 layers.

## 3.1 LANGUAGE MODELING

**Training Curves.** Figure 2 presents validation BPB metrics throughout training for both Large and XL model scales. At the *Large scale*, we make note of the following comparisons: (i) all isotropic models severely underperform hierarchical models. Among these, **MambaByte** is significantly better than **LlamaByte**, both the FLOP-matched sliding window attention (SWA) variant and even the global attention variant that is data-matched but uses $2\times$ the FLOPs; (ii) **H-Net (pool)** is much worse than all other H-Net variants, validating that fixed-width chunking is not effective; (iii) **SpaceByte** is much worse than **SpaceByte++**, validating our strategy for network design as well as usage of Mamba in the outer networks (Section 2.1); (iv) **SpaceByte++** is in turn worse than **H-Net (space)**, validating our improved signal propagation techniques (Section C.3); (v) **H-Net (space)** is a very strong model reaching the performance of the **BPE Transformer**, validating the effect of data-dependent chunking strategies together with a well-designed hierarchical architecture; (vi) **H-Net (1-stage)** is stronger than **H-Net (space)**, validating that our dynamic chunking mechanism successfully learns how to segment data in a *context-dependent* way that improves over strong heuristics; and (vii) **H-Net (2-stage)** is significantly better than **H-Net (1-stage)**, validating that iterated dynamic chunking can potentially learn a nested hierarchy of useful features, and leverage compute and parameters even more effectively.

At the *XL scale*, we zoom in more closely and compare only the strongest set of methods: SpaceByte++, H-Net (space), H-Net (1-stage), and H-Net (2-stage). The same trends hold as at the Large scale. Our SpaceByte++ baseline is strong, but slightly worse than the BPE Transformer baseline. On the other hand, **all byte-level H-Net methods start off worse than the token-level Transformer, but scale better after enough data.** H-Net (space), H-Net (1-stage), and H-Net (2-stage)

Table 2: **Zero-shot performance comparison across multiple benchmarks, all data-/FLOP-matched.** Evaluation results on seven downstream tasks at both Large (760M) and XL (1.3B) scales. GFLOPS/BYTE is measured on FineWeb-Edu validation set, reported as the average over the course of training. See Table 1 for architectural details.

| MODEL | INPUT | GFLOPS/BYTE | F-EDU BPB↓ | LMB. ACC↑ | HELLA. ACC_n↑ | PIQA ACC↑ | ARC-E ACC↑ | ARC-C ACC_n↑ | WINO. ACC↑ | OPEN. ACC_n↑ | AVERAGE ACC↑ |
|---|---|---|---|---|---|---|---|---|---|---|---|
| **#FLOPs matched to GPT-3 *Large*** | | | | | | | | | | | |
| Transformer | Token | 0.42 | 0.756 | 45.0 | 54.5 | 72.3 | 69.9 | 36.3 | 55.9 | 38.8 | 53.3 |
| LlamaByte | | 0.42 | 0.859 | 37.0 | 40.5 | 64.7 | 55.1 | 26.7 | 52.3 | 32.4 | 44.1 |
| LlamaByte (Global) | | 0.95 | 0.845 | 36.4 | 41.5 | 65.7 | 57.2 | 27.1 | 49.8 | 32.2 | 44.3 |
| MambaByte | | 0.42 | 0.845 | 32.9 | 42.0 | 66.2 | 55.9 | 28.1 | 51.7 | 33.2 | 44.3 |
| SpaceByte | | 0.41 | 0.791 | 43.0 | 49.0 | 69.0 | 63.3 | 33.5 | 53.3 | 35.0 | 49.4 |
| SpaceByte++ | Byte | 0.42 | 0.760 | **48.0** | 55.7 | 71.3 | 67.9 | 35.4 | 57.5 | 39.6 | 53.6 |
| H-Net (pool) | | 0.42 | 0.780 | 43.2 | 54.7 | 69.7 | 67.9 | 34.7 | 54.8 | 36.4 | 51.6 |
| H-Net (space) | | 0.42 | 0.755 | 46.7 | 55.9 | 72.4 | 68.8 | 34.6 | 57.6 | 38.0 | 53.4 |
| H-Net (1-stage) | | 0.43 | 0.755 | 46.2 | 55.5 | 71.0 | 68.1 | 35.6 | 58.6 | 40.0 | 53.6 |
| H-Net (2-stage) | | 0.43 | **0.743** | 46.9 | **57.4** | 72.0 | **71.7** | 39.2 | 60.4 | 40.6 | **55.5** |
| **#FLOPs matched to GPT-3 *XL*** | | | | | | | | | | | |
| Transformer | Token | 0.69 | 0.730 | 48.1 | 58.0 | 73.1 | 72.2 | 37.5 | 58.6 | 40.8 | 55.5 |
| SpaceByte++ | | 0.72 | 0.733 | **51.3** | 60.1 | 72.4 | 71.8 | 38.0 | 58.5 | 40.6 | 56.1 |
| H-Net (space) | Byte | 0.70 | 0.726 | 50.3 | 61.5 | 73.6 | 72.4 | 40.2 | 60.2 | 41.8 | 57.1 |
| H-Net (1-stage) | | 0.72 | 0.728 | 48.4 | 59.5 | 72.4 | 73.0 | 38.3 | 59.2 | 42.4 | 56.2 |
| H-Net (2-stage) | | 0.69 | **0.715** | 50.5 | 62.2 | 73.7 | 74.2 | 42.2 | 60.5 | 44.0 | 58.2 |

Table 3: **Robustness evaluation on HellaSwag with textual perturbations, all data-/FLOP-matched.** Zero-shot accuracy on five different perturbation types (AntSpeak, Drop, RandomCase, Repeat, UpperCase) for models trained exclusively on clean data without noise augmentation. Best and second best results in each column are denoted using bolded and underlined texts, respectively. The Robustness Score metric show that all byte-level models are more robust to adversarial text inputs than tokenizer-based Transformer. H-Net (2-stage) shows significantly enhanced robustness in textual perturbations, with the highest average accuracy across all noise types and highest robustness score. See Table 1 for architectural details, and Section D.2 for the definition of Robustness Score.

| MODEL | INPUT | HELLASWAG | | | | | AVERAGE↑ | ROBUSTNESS SCORE↑ |
|---|---|---|---|---|---|---|---|---|
| | | ANTSPEAK | DROP | RANDOMCASE | REPEAT | UPPERCASE | | |
| **#FLOPs matched to GPT-3 *Large*** | | | | | | | | |
| Transformer | Token | 31.1 | 29.9 | 27.1 | 27.8 | 38.9 | 30.9 | 20.2 |
| LlamaByte (W1024) | | 30.4 | 28.1 | 29.3 | 27.2 | 38.5 | 30.7 | 36.9 |
| LlamaByte (Global) | | 31.1 | 28.1 | 29.7 | 27.3 | 39.0 | 31.0 | 36.6 |
| MambaByte | | 29.8 | 27.9 | 29.9 | 27.1 | 39.6 | 30.9 | 34.5 |
| SpaceByte | | 30.7 | 29.8 | 33.5 | 29.5 | 47.8 | 34.3 | 38.1 |
| SpaceByte++ | Byte | 31.0 | 30.9 | 35.8 | 29.3 | 54.0 | 36.2 | 36.4 |
| H-Net (pool) | | 30.5 | 31.2 | 35.4 | 29.6 | 53.4 | 36.1 | 37.3 |
| H-Net (space) | | 30.8 | 31.2 | 38.6 | 29.4 | 54.0 | 36.8 | 38.2 |
| H-Net (1-stage) | | **31.2** | 31.1 | 35.4 | 29.9 | 54.1 | 36.4 | 37.2 |
| H-Net (2-stage) | | 30.8 | 32.1 | **39.3** | 30.4 | **57.1** | **38.0** | **39.0** |
| **#FLOPs matched to GPT-3 *XL*** | | | | | | | | |
| Transformer | Token | **31.6** | 30.7 | 28.0 | 28.5 | 43.0 | 32.3 | 22.2 |
| SpaceByte++ | | 30.9 | 32.1 | 40.3 | 30.6 | 58.5 | 38.5 | 38.5 |
| H-Net (space) | Byte | 31.2 | 33.2 | 41.9 | 31.8 | 60.7 | 39.8 | 40.5 |
| H-Net (1-stage) | | 30.9 | 32.7 | 39.2 | 31.2 | 58.4 | 38.6 | 39.5 |
| H-Net (2-stage) | | 31.1 | **34.7** | **44.1** | **33.0** | **61.7** | **40.9** | **42.8** |

cross over the tokenized Transformer after just 200B bytes, 100B bytes, and 30B bytes respectively. Beyond these points, H-Net's performance advantage widens progressively, demonstrating that the benefits of learnable dynamic chunking get strengthened with additional training data, as the model continuously refines its chunking strategy.

**Downstream Evaluations.** Table 2 presents zero-shot accuracy across diverse downstream benchmarks (Paperno et al., 2016; Zellers et al., 2019; Bisk et al., 2020; Clark et al., 2018; Sakaguchi et al., 2021; Mihaylov et al., 2018) using `lm-evaluation-harness` (Gao et al., 2024) for models at Large and XL scales. SpaceByte++, H-Net (space), and H-Net (1-stage) all have similar performance to the BPE Transformer at *Large* scale, and slightly outperform it at the *XL* scale, consistent with their close training curves (and possibly reflecting some noise in the evaluations). **H-Net (2-stage) consistently achieves the highest performance across most tasks**, outperforming 2.2% and 2.6% over the Transformer baseline at *Large* and *XL* scales respectively. Notably, the *Large* H-Net (2-stage) matches the average downstream performance of the *XL* BPE Transformer.

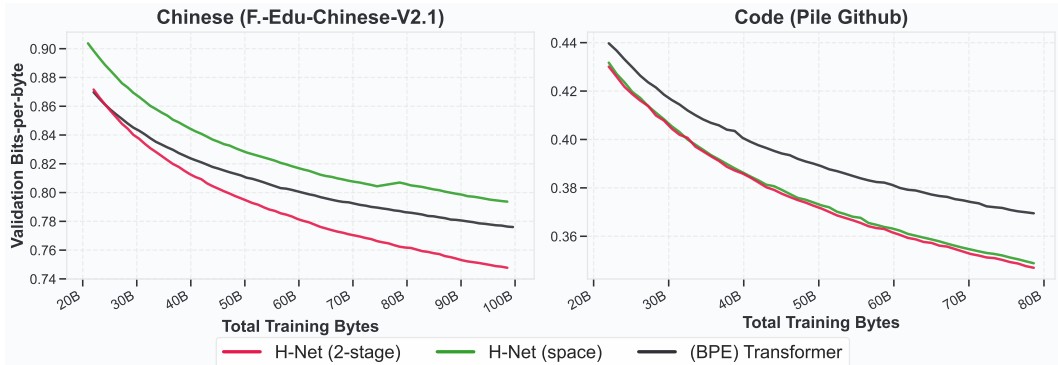

Figure 3: **Validation Bits-per-byte (BPB) throughout training on Chinese language and code modeling.** H-Net (space) and H-Net (2-stage) are byte-level, while the Transformers use the Llama-3 tokenizer which was designed for multilingual. H-Net clearly outperforms both Transformer and H-Net (space) on Chinese language modeling, which does not have space-like segmentation cues, with lower BPB than H-Net (space) throughout training and crossing over with Transformer after around 25B bytes. On code, both H-Net (2-stage) and H-Net (space) significantly outperform BPE Transformer. Final post-decay results can be found in Table 5.

**Robustness to Textual Perturbations.** Table 3 evaluates model robustness on HellaSwag with various textual perturbations, following protocols from BLT (Pagnoni et al., 2024). Importantly, these are the same checkpoints trained on clean FineWeb-Edu data used to evaluate Table 2), without any form of special data mix or augmentations that may improve character-level robustness. H-Net (2-stage) demonstrates substantially improved robustness compared to all baselines, with performance gaps exceeding those observed in standard benchmarks.

## 3.2 ALTERNATE LANGUAGE DATASETS

Besides conventional language modeling, we also examine three other language modeling settings – Chinese, code, and DNA. These three settings present distinct challenges for traditional language-modeling pipelines: (i) Chinese characters consist of 3 utf-8 encoded bytes each and Chinese language does not have natural spaces; thus, constructing a vocabulary or picking boundaries requires special consideration; (ii) Code contains much more whitespace than typical language, which allows greater compressibility if handled properly, and it also has latent hierarchical structure that can be leveraged for improved reasoning capabilities; and (iii) DNA does not have any natural tokenization cues and instead must be processed as raw base pairs. In contrast, H-Net can operate on raw data without the need for handcrafted features (whether vocabulary or delineation cues); it therefore provides a natural architecture that can operate naturally on any language.

As demonstrated in Figure 3, we find that H-Net (2-stage) scales better than BPE Transformer (with the Llama3 tokenizer) and H-Net (space) on both Chinese and code, and achieves lower compression after the decay phase (see Table 5). We additionally measure the performance of each Chinese-language model on the Chinese split of XWinograd, a multilingual Winograd Schema Challenge (Muennighoff et al., 2023), where H-Net (2-stage) is significantly better than H-Net (space) which in turn is better than Transformer as shown in Table 5.

## 4 CONCLUSION

Major advances in deep learning have resulted from powerful architectural innovations enabling previously-handcrafted features to be learned from data, from CNNs learning visual features (Krizhevsky et al., 2012) to Transformers discovering linguistic patterns (Vaswani et al., 2017). **H-Nets** similarly unlock the ability to remove another layer of pre-processing, such as tokenizers, and instead learn them end-to-end. This ability results from a set of new techniques we introduce that work together to form a **dynamic chunking** mechanism, which is able to learn content- and context- dependent discrete segmentation strategies through standard gradient-based optimization. A single-stage byte-level H-Net already exceeds the performance of standard tokenized language models, and recursive H-Nets with multiple stages of dynamic chunking further improve its scaling. H-Nets substantially remedy issues with tokenizers, display very strong performance on diverse languages and language-like modalities, and more broadly may serve as the backbone of general foundation models that do *more learning* with *less processing*.

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

APPENDIX

# A DISCUSSION

**Distillation.** For new architectures, showing that they can be distilled from standard pretrained Transformers can result in stronger new models with substantially reduced training (Bick et al., 2024). In Section E.2, we investigate this for H-Net by initializing the main network from a pre-trained Llama checkpoint and learning the encoder and decoder networks. With less than 200B bytes of training, the resulting model shows strong performance much better than if it were trained from scratch, although still worse than the teacher model. Our distillation procedure is perhaps currently the most efficient way of creating an end-to-end byte-level model, but we expect that it can be further improved.

**Efficiency.** Because of the dynamic nature of our model, it requires different considerations in making both the training pass and inference step efficient. Our implementation incorporates several engineering techniques already, such as handling variable sequence lengths within a mini-batch using specialized kernels provided by Dao (2024); Dao & Gu (2024). Because of the different architectural considerations, it is difficult to compare to more standard pipelines; our current implementation may be approximately up to $2\times$ slower than an isotropic model during training.

Note that the memory usage of our model is also dynamic, unlike standard sequence models, so other edge cases may happen, such as unlucky batches of sequences that are too long and overflow the device memory. Relatedly, one difficulty with stepping H-Net in batched mode is that different tokens in the batch may require different amounts of compute.

We believe that such considerations are not fundamental and will be an important subject of future work; just as how related dynamic sparsity and conditional compute methods such as Mixture-of-Experts and speculative decoding (Leviathan et al., 2023; Chen et al., 2023) benefited from years of dedicated engineering improvements.

**Deeper Hierarchies.** H-Net is the first *dynamic* hierarchical model that can *recursively* nest its chunking strategy (see Table 4 and Section B). In this paper, we showed that iterating H-Net from 0 stages (i.e. an isotropic model) to 1 stage and from 1 stage to 2 stages consistently improves performance. We did not attempt a 3-stage H-Net at all for simplicity. Testing if H-Net can be iterated even deeper remains an immediate direction to explore.

**Global Sequence Model Considerations.** Much research on sequence model architectures has focused on individual layers, where the tradeoffs are often quite direct. For example, recurrent models such as state space models (Gu, 2023; Gu & Dao, 2024) and linear attention variants (Katharopoulos et al., 2020; Yang et al., 2024a;b; 2025) compress arbitrarily long sequences into fixed-size hidden states, offering higher efficiency at the expense of precise retrieval of information (e.g. struggling with recall (Jelassi et al., 2024)).

H-Net, however, is a *global* architectural design that is simultaneously orthogonal to, but may have interactive effects with, the choice of individual layers. For example, using deeper hierarchies with exclusively recurrent layers would preserve linear computation (in sequence length) but *logarithmic* state size, resembling newer sequence model layers such as log-linear attention (Guo et al., 2025) and Prefix Scannable Models (Yau et al., 2025), but with dynamic hierarchies. Similarly, the recursive compression of sequence length may alleviate their limitations in retrieval on long sequences. This may be considered a form of *dynamic state allocation*. This paper has not focused on such implications, which would be a possible direction for future research.

**Long Context.** Similarly, an effect of the global hierarchical structure may be improved long context abilities, which is a common motivation for hierarchical models (Koutnik et al., 2014; Chang et al., 2017). Much research on sequence models again focuses on long context at the layer level (Poli et al., 2023; Gu & Dao, 2024; Vaswani et al., 2017), and we hypothesize that H-Nets may provide general long context improvements in an orthogonal direction.

**Latent Test-Time Compute.** Test-time compute techniques, exemplified by Chain-of-Thought (Wei et al., 2022), have been shown to improve model performance on a variety of reasoning benchmarks (Muennighoff et al., 2025; OpenAI, 2024). Recent work has explored including latent representations (as opposed to just tokens) in the reasoning process (Hao et al., 2024), culminating in "recurrent depth" models that roll out an RNN for as many steps as needed before emitting a token (Geiping et al., 2025). As discussed in Section C.4, H-Net is also capable of dynamically changing compute per output generated; thus, it can be viewed as a model that can dynamically allocate

latent test-time compute as well. Additionally, as the motivation of H-Net is to recursively build higher-order abstractions, we hypothesize that it would be more effective as a reasoning model that operates over its own learned concepts instead of arbitrary token inputs.

**Sparsity.** H-Net can be viewed as a form of dynamic sparsity or conditional computation, and is related to concepts such as mixture-of-experts (MoE) (Fedus et al., 2022; Shazeer et al., 2017) and mixture-of-depths (Raposo et al., 2024). We showed that at the byte level, DC is much more effective than MoE when controlled for parameters and compute (Figure 14), and leave fleshing out further connections and comparisons for future work. We also note that H-Net can be viewed as orthogonal to MoE, which can be applied to sparsify any MLP layers within an H-Net.

**Scale.** The largest models in this paper were FLOP-matched to the equivalent of a 1.3B parameter Transformer. While we believe that this provides sufficient evidence for the effectiveness of this approach, it remains to validate H-Net at larger model sizes of 3B, 7B, and beyond. We note that while we observed no instabilities at our model sizes, the added complexity of H-Net and inherent difficulties of learning end-to-end discrete selection problems may require more serious investigation of potential stability challenges at larger scale.

**Scaling Laws.** Formally estimating the scaling behavior of a model requires calculating scaling law coefficients that sweep across a large range of model sizes and compute horizons (Kaplan et al., 2020; Hoffmann et al., 2022). We did not pursue this formal approach in this paper due to resource constraints.

Instead, we used a simpler heuristic for the scaling behavior of our models, at least with respect to data. We note that

- essentially all modern models live in the "overtrained" regime (with respect to the formal scaling laws) due to inference considerations at deployment (Touvron et al., 2023b); and
- these overtrained models often use modern schedulers that have extended periods of constant learning rates (Hu et al., 2024; DeepSeek-AI, 2024).

Thus, we decided to use the models' losses during the constant phase as a proxy for how quickly they improve with data. We believe this still provides useful insight into scaling behaviors, and a more dedicated analysis of formal scaling laws remains an important topic for future work.

**BPB Calculation.** For baseline BPE tokenized models throughout this work, we used the standard bits-per-byte (BPB) calculation of simply rescaling the negative log-likelihood (or log perplexity) by the average number of bytes per token (Gao et al., 2020; Wang et al., 2024; Slagle, 2024). However, this is not strictly speaking a correct BPB estimate for tokenized models, as it assumes that the probability the model outputs a string is equal to the probability of the model outputting the greedy tokenization of the string.

Depending on how the model is trained, it is possible the model can output other tokenization sequences with nonzero probability. There are an exponential number of these, so computing the exact BPB is intractable; however, concurrent work (Vieira et al., 2024) shows that the standard BPB calculation indeed overestimates BPB. Due to the high computational overhead of estimating the true BPB, we only provide the standard (inexact) value; nevertheless, H-Net's superior performance on downstreams provides supporting evidence that it scales better than BPE models.

Table 4: **Related architectures.** Comparison of related architectures, particularly those focused on byte-level modeling. H-Net is the first architecture that enables dynamic, multi-stage hierarchies. Extended discussion is provided in Section B.

| CLASS | AUTOREGRESSIVE | CHUNKING MECHANISM | MULTI-STAGE HIERARCHY | EXAMPLE ARCHITECTURES |
|---|---|---|---|---|
| Isotropic | ✗ | — | — | ByT5 |
| | ✓ | — | — | MambaByte |
| Hierarchical (static) | ✗ | $k$-width pooling | ✓ | Funnel-Transformer
Canine
Charformer |
| | ✓ | $k$-width pooling | ✓ | Hourglass Transformer
SaShiMi
MegaByte
Block Transformer
MBLM
AU-Net 3 |
| Hierarchical (external) | ✗ | delimiters | ✗ | eByte
WSF |
| | ✓ | delimiters | ✗ | DPT (Whitespaces)
SpaceByte
AU-Net 2 |
| | | entropy | ✗ | DPT (Entropy)
BLT |
| Hierarchical (dynamic) | ✓ | soft matching | ✗ | MANTa |
| | | soft gating | ✗ | MrT5 |
| | | stochastic reparameterization | ✗ | DPT (Gumbel) |
| **Hierarchical (dynamic)** | ✓ | **dynamic chunking** | ✓ | **H-Net** |

## B   RELATED WORK

The fundamental challenge of transforming raw sequential data into computationally efficient representations manifests across multiple domains through implicit chunking processes. In language modeling, this challenge is addressed through tokenization using static vocabularies derived from frequency-based algorithms such as Byte-Pair Encoding (BPE) (Sennrich et al., 2015) in GPT models (Radford et al., 2019; Brown et al., 2020) and SentencePiece (Kudo & Richardson, 2018) in Llama architectures (Touvron et al., 2023b; Grattafiori et al., 2024). Computer vision addresses similar challenges through spatial pooling operations (Ronneberger et al., 2015) that aggregate neighboring pixels into meaningful representations.

Despite achieving strong empirical performance, it is widely known that traditional tokenization approaches in language models suffer from fundamental limitations that constrain model capabilities. Fixed vocabularies exhibit biases toward high-resource languages, demonstrate fragility when handling adversarial inputs, and show lower performance on character-level tasks (Petrov et al., 2023; Ahia et al., 2023; Belinkov & Bisk, 2017; Sun et al., 2020; Xue et al., 2022). These limitations stem from the static nature of predefined vocabularies, which cannot adapt their chunking strategies to input content or context.

To address these constraints, *tokenizer-free* methods have emerged that avoid the reliance on predefined vocabularies.

- In Section B.1, we discuss the most directly related prior work on autoregressive sequence models, extending the overview from Section 1.

- In Section B.2, we discuss non-autoregressive models. We note that essentially all autoregressive architectures can be turned into non-autoregressive architectures (including our proposed H-Net), and vice versa, which provide possible extensions of H-Net in future work. However, we provide this delineation because it marks an important difference in motivation that influences design considerations and downstream evaluations.

- Section B.3 mentions other works in non-language modalities related to tokenization.

We summarize our discussion on tokenizer-free architectures in Table 4.

### B.1   AUTOREGRESSIVE TOKENIZER-FREE ARCHITECTURES

As outlined in Section 1, prior work on autoregressive tokenizers for architectures can be divided into four categories:

1. Non-hierarchical *isotropic* architectures.

2. Hierarchical architectures with *static* chunking strategies, where chunk boundaries are content-agnostic (usually some variant of fixed-width pooling).

3. Hierarchical architectures with *external* chunking strategeies, where chunk boundaries are provided by an external function or module.

4. Hierarchical architectures with *dynamic* chunking strategies, where chunk boundaries are content-dependent and learned end-to-end.

#### B.1.1   ISOTROPIC ARCHITECTURES

The most direct approach to modeling language with tokenizers is to simply model raw byte sequences with a standard sequence model architecture. Since this naive approach suffers from computational challenges on long sequences, MambaByte (Wang et al., 2024) proposed using a state space model for its linear-time efficiency. We similarly use Mamba(-2) (Dao & Gu, 2024) layers in the outer stages of an H-Net. Notably, through extensive ablations we show that Mamba is not just more efficient but also better at modeling high-resolution data such as text characters and DNA base pairs.

#### B.1.2   STATIC CHUNKING

To reduce sequence length, several approaches downsample the input sequence hierarchically. The most straightforward methods operate independently of input context, partitioning sequences using fixed-size intervals. Many strategies could be used to aggregate a width-$k$ window, including direct

downsampling, average pooling, linear transformations that mix across the chunk, convolutions, and more; we lump these together as *pooling* operations.

Hourglass Transformer (Nawrot et al., 2022) and MegaByte (Yu et al., 2023) exemplify this strategy. Other recent variants include the Block Transformer (Ho et al., 2024) and Multiscale Byte Language Model (MBLM) (Egli et al., 2025), which use similar multi-stage static chunking architectures. Concurrently to H-Net, the MBLM also proposes using Mamba layers in the outer stages.

These approaches share conceptual similarity with spatial pooling operations in vision models that reduce resolution through fixed-window aggregation (Krizhevsky et al., 2012; He et al., 2016). While these content-agnostic methods have simple and efficient implementations, they face an inherent limitation: they do not reflect natural semantic boundaries in the data. Fixed-size chunking inevitably creates arbitrary separations that can split meaningful units such as words, morphemes, or phrases, thereby limiting model expressivity.

This class of models may also be called "autoregressive U-Nets", characterized by the U-Net multiscale architecture (Ronneberger et al., 2015) with additional considerations to maintain causality. Prior to these, the S4 and SaShiMi models (Gu et al., 2022; Goel et al., 2022) used the same architecture successfully in the vision and audio modalities, where fixed-window downsampling exhibits more appropriate inductive bias in contrast to language. SaShiMi specifically operated over 8-bit quantized audio inputs, hence also was a form of byte-level modeling that used BPB as a metric.

### B.1.3    EXTERNAL CHUNKING

An improvement to hierarchical architectures with static downsampling is to use content-aware chunking strategies that attempt to identify natural token boundaries based on semantic or statistical properties of the input data. Several recent models propose using the boundaries provided by an external module, with two main variations appearing.

**Delimiter-based methods.**    The most intuitive content-aware approach segments on surface-level syntactical boundaries, which can be often implemented by simple rules or regular expressions.

Dynamic Pooling Transformer (DPT) (Nawrot et al., 2023) proposed a variant that segmented on whitespace characters, effectively making each word its own token. SpaceByte (Slagle, 2024) extends this to "space-like" delimiters (*e.g.,* /, ], :) as natural boundary signals. This approach provides semantically meaningful chunking for languages with explicit word separators such as English text and code.

However, delimiter-based methods cannot be used for inputs lacking explicit separators (e.g. many non-European languages, or other modalities such as DNA). Additionally, these approaches cannot be extended to multi-level hierarchical chunking due to ambiguities in defining natural delimiters at higher semantic levels. AU-Net (Videau et al., 2025) is a concurrent work that augments SpaceByte with additional stages of hierarchy using fixed-width chunking. Specifically, AU-Net 2 is SpaceByte with minor architectural modifications, while AU-Net 3 (and AU-Net 4) add additional levels of hierarchical with width-2 downsampling.

In this work, we show that SpaceByte's delimiter chunking strategy can be a very powerful baseline on appropriate languages – competitive with or outperforming traditional tokenizers on English and code – when augmented with several of H-Net's additional techniques (Section 3.1, Section E.3, Figure 3, Figure 9).

**Entropy-based methods.**    Another approach to circumvent the delimiter dependency is using the autoregressive conditional entropy as a heuristic to identify semantic boundaries. This was first proposed by the Dynamic Pooling Transformer (DPT) (Nawrot et al., 2023), which detects entropy spikes that correlate with semantic transitions. The recent Byte Latent Transformer (BLT) (Pagnoni et al., 2024) employs entropy thresholds computed by a separate pre-trained model to determine chunking boundaries.

Despite showing promise, these entropy-based approaches face several practical limitations. First, they require extensive domain-specific hyperparameter tuning to establish appropriate entropy thresholds, reducing their general applicability. Second, they still fall behind in performance; for example, BLT necessitates an extra 3B parameters (at the 8B scale) solely for multi-gram hash embeddings to match BPE Transformer baselines. Finally, these methods also cannot be extended

hierarchically because computing cross-entropy loss requires access to target vocabularies, which are unavailable for intermediate latent representations in multi-stage architectures.

In this work, we do not compare against BLT because of its complexity: (i) necessitating training an auxiliary language model to provide proxy autoregressive conditional entropies (ii) converting it into an external neural tokenizer through tuning entropy heuristics (iii) using hash embeddings, which can be considered an orthogonal architectural component which may be incorporated into H-Net as well if desired.

Instead, we compared against SpaceByte (and our own stronger versions of SpaceByte), which we believe to be representative of the external-chunking family of methods and competitive to the entropy-based chunking strategy of BLT (for our main experiments such as English data).

### B.1.4 DYNAMIC CHUNKING

The ideal tokenizer-free architecture would incorporate a *dynamic chunking method* that attempts to learn optimal segmentation strategies directly from data through gradient-based optimization. Such a method would be optimized jointly together with the outer (fine-resolution) and inner (coarse-resolution) networks, and be able to create boundaries that are *content-* and *context-* aware.

The only prior work we are aware of that attempted a true dynamic chunking method is (one variant of) the Dynamic Pooling Transformer (DPT) (Nawrot et al., 2023), which incorporates stochastic exploration mechanisms using Gumbel noise (Jang et al., 2017; Maddison et al., 2017) to enable differentiable boundary selection during training. Despite their theoretical flexibility, trainable methods encounter critical challenges. The stochastic exploration process requires careful tuning of noise magnitudes and introduces high-variance gradients that destabilize training, making it difficult to scale to larger model sizes.

In practice, the end-to-end (stochastic reparameterization) variant of DPT underperformed the external chunking variants (drawing boundaries on entropy spikes or whitespaces) (Nawrot et al., 2023), illustrating the difficulty of this problem. Furthermore, the training instability prevented DPT from expanding to multiple hierarchical stages, constraining these methods to single-stage chunking.

We additionally highlight simple architectural modifications of DPT motivated by improved inference (Fleshman & Van Durme, 2023) or multilingual ability (Ahia et al., 2024). Such techniques can also be easily adapted to H-Nets in future work.

### B.2 NON-AUTOREGRESSIVE TOKENIZER-FREE ARCHITECTURES

Each class of autoregressive architectures from Section B.1 has corresponding non-autoregressive variants as well. Although these often have similar design principles, they are also motivated by different tasks, settings, and design considerations (e.g. no evaluation on large-scale autoregressive pretraining) and thus can be difficult to compare directly to autoregressive models. We include these for context and completeness.

**Isotropic.** ByT5 (Xue et al., 2022) directly models bytes using a bidirectional encoder-decoder architecture, showing improved performance with small models (because more power is moved into model parameters rather than vocabulary embeddings) and spelling-sensitive tasks.

**Hierarchical (Static).** Funnel-Transformer (Dai et al., 2020) is an early architecture that uses a U-Net-like architecture for language, focusing on the non-causal setting. Canine (Clark et al., 2022) proposes a hierarchical model with convolution-based static downsampling; their method also targets non-autoregressive language models.

Charformer (Tay et al., 2021) presents a gradient-based subword tokenization (GBST) method that pools the input sequence at different resolutions, inducing an implicit ensemble of hierarchical models. It shows improved efficiency to performance trade-offs compared to models that use a single downsample resolution.

We note that these methods can also be endowed with implicit supervision from external tokenizers; for example, Canine proposes a variant that uses subword tokens in the *objective function* (via masking out subwords in the masked language modeling objective), but does not need the tokenizer at inference time. We also note that such techniques are particular to non-autoregressive models, since they allow for variations in the modeling objective.

**Hierarchical (External).**    Thawani et al. (2023) propose the eByte method, which resembles MegaByte but chunks on spaces with Transformer-based CLS-token pooling, and lacks the byte-level residual stream that enables autoregressive modeling.    Word-based self-attention fusion (WSF) (Sreedhar et al., 2023) proposes a similar pooling strategy for encoder language models.

**Hierarchical (Dynamic).**    MANTa (Godey et al., 2022) introduces an end-to-end method that predicts segmentation boundaries and pools bytes into blocks using a matching objective. MrT5 (Kallini et al., 2025) is a recent method improving on ByT5 with a gating mechanism that allows for explicit dynamic token-merging at inference time, reducing sequence lengths by up to 80%.

### B.3   OTHER TOKENIZATION-RELATED WORK

**Tokenizers for Other Modalities.**    While computer vision pipelines do not use tokenizers like BPE in the same way as language models do, they frequently need to turn raw perceptual data (images and videos) into shorter sequences of representations. One approach is the simple patchification step first introduced by the Vision Transformer (ViT) (Dosovitskiy et al., 2021). However, images, videos, and audio can have varying amounts of semantic content and non-uniform redundancies. A number of more recent approaches attempt to produce variable length tokenizations that adapt to the information content of the data, Which performs a more similar role to tokenization in language models.  This can be done in the latent space of an autoencoder (Yu et al., 2024; Duggal et al., 2024) or through explicit token merging (or "run length encoding") with heuristics (Bolya et al., 2022; Choudhury et al., 2024).  In the audio domain, SlowAE (Dieleman et al., 2021) proposes a joint autoencoder with autoregressive modeling that finds semantic segmentation boundaries, which resembles H-Net's approach at a high level.

FAST (Lin et al., 2025) introduces a tokenizer for robotics, Which tokenizes continuous control actions by combining the Discrete Cosine Transform (DCT) with BPE.

**Vocabulary Scaling.**    While scaling laws for language models have generally kept tokenizers fixed (Kaplan et al., 2020; Hoffmann et al., 2022; Grattafiori et al., 2024), recent works have showed that the tokenizer also warps scaling laws, in fact more so than model architecture changes (Mayilvahanan et al., 2025). Tao et al. (2024) and Huang et al. (2025) directly show that it is more optimal to scale an LLM's vocabulary together with the rest of the model parameters.

In H-Nets, which are designed to operate over higher resolution raw data, the actual vocabulary can be kept minimal, but the chunking mechanism can be viewed as an implicit "tokenizer" with infinite vocabulary.  As H-Nets scale in size, one expects that more iterations of hierarchy can be added (increasing effective chunk size), or the chunk size can directly be increased to leverage parameters more efficiently.  This resembles the idea of increasing a vocabulary in tokenized models (which would generally increase the average length of tokens).

SuperBPE (Liu et al., 2025) shows that allowing vocabulary tokens to cross whitespace boundaries can also improve performance.  This is related to H-Net's motivation of higher-level chunking of words into phrases; empirically, Figure 11 shows how the 2-stage H-Net finds semantic multi-word groups in the inner stage.

**Cross-Tokenizer Transfer.**    Minixhofer et al. (2024) and Minixhofer et al. (2025) address the problem of *tokenizer transfer*, or adapting models across different tokenizers (for example for cross-language or cross-modality usage, or for knowledge distillation).

**Other Effects of Tokenization.**    Lee et al. (2024) discuss the effects that tokenization has on arithmetic in LLMs. For example, comparing the performance of left-to-right vs. right-to-left tokenization. Hayase et al. (2024) show that examining the vocabulary of a BPE tokenizer leaks information about the data mix that it was trained on.

**Tokenization Theory.**    Schmidt et al. (2024) examined the hypothesis that the primary role of tokenization is to shrink the input sequence length.  They invented a new tokenizer that has even higher compression rates than BPE (actually, they keep the same vocabulary but simply find different segmentations that are more compressed) yet leads to worse language models, providing evidence against the hypothesis.

Rajaraman et al. (2024) showed that for certain data distributions, applying tokenization qualitatively changes what Transformers can learn.

Phan et al. (2024) and Vieira et al. (2024) propose various algorithms for converting a language model over tokens into a language model over characters or bytes. This helps alleviate some limitations of tokenizers such as the "prompt boundary" problem, the ability to compare different LLMs with different tokenizers, and simply produces better estimates of a language model's true compressive ability (as measured by bits-per-byte). However, such algorithms are complex and expensive, and compared to direct byte-level models they are not practical for use during inference decoding (repeated autoregressive sampling).

## C  MODEL DETAILS

### C.1  DESIGN PRINCIPLES

**Encoder and Decoder Networks.**    The encoder and decoder networks in H-Net face unique design constraints due to their dual objectives and computational requirements. Each encoder must simultaneously (i) preserve fine-grained information for transmission to its corresponding decoder through residual connections equation 3, and (ii) compress inputs into chunks of richer representations for the main network. The decoder, in turn, must effectively combine coarse-grained representations from the main network with fine-grained details from the encoder residuals.

Importantly, both encoders and decoders operate on uncompressed sequences, making computational efficiency a significant design constraint that shapes our architectural choices. Recent studies demonstrate that state space models (SSMs) (Gu et al., 2022; Gu & Dao, 2024) excel at processing fine-grained data including audio (Goel et al., 2022), DNA sequences (Schiff et al., 2024), and robotic control signals (Lu et al., 2023).

Based on these insights, we employ Mamba-2 layers (Dao & Gu, 2024) as the primary building blocks for the encoder and decoder networks. This choice yields two significant benefits: effective handling of fine-grained inputs, and substantially improved efficiency when processing long, uncompressed sequences. Our ablation studies (Section E.3) confirm that SSM-based encoders/decoders significantly outperform Transformer layers, not just at the byte level but even on coarser inputs, which we attribute to their stronger inductive bias for compression which helps build abstractions (Gu, 2025).

**Main Network.**    H-Net's computational efficiency stems from strategic parameter allocation. We concentrate the majority of model capacity in the main network, which operates on progressively compressed sequences. After $S$ stages of compression, the main network receives sequences where $L^S \ll L^0$, enabling much larger networks within the same computational budget. This design reflects two key principles: (i) compressed sequences allow more parameters and compute per chunk, and (ii) higher-level abstractions benefit from increased processing power.

The main network functions as a standard language model and can employ any sequence mixing architecture. We default to Transformer layers for two reasons: compressed representations align well with Transformers' strengths in processing discrete, semantically-rich tokens, and this choice enables more controlled comparison with traditional BPE-based Transformer baselines in our experiments. However, the modular design also allows straightforward substitution with alternative architectures (*e.g.,* a state space model, hybrid, or H-Net itself) as explored in our ablations.

**Architectural Guidelines.**    Compared to standard isotropic models, the H-Net's structure introduces several new dimensions of architectural parameters to balance the parameter/compute allocation to each network. To simplify the search space, we follow a few general guidelines.

- First, we ensure the model width (often referred to as $d_{\text{model}}$ for isotropic architectures) is monotone in the hierarchy: $D^0 \leq D^1 \leq \cdots \leq D^S$. This allows increasing compute and parameters used in the main network without significantly increasing its depth.
- Second, using efficient and powerful SSM layers in the outer networks allow reducing the number of layers used compared to similar prior architectures that only used Transformer layers (Slagle, 2024); in this paper, we always stick to four layers (or the equivalent of four Mamba layers) in each encoder/decoder network.

To handle the changes in dimensions without an additional linear layer, we adopt the technique used in SpaceByte (Slagle, 2024) with the marginal change: to expand dimensions (*i.e.,* $D^s \to D^{s+1}$), we append all vectors with a shared trainable vector of dimension $D^{s+1} - D^s$; to reduce dimensions (*i.e.,* $D^{s+1} \to D^s$), we take the first $D^s$ dimensions from each vector.

We note that H-Net's performance can likely be improved with more careful tuning of the layer allocation and hyperparameters between sub-networks.

### C.2  DYNAMIC CHUNKING (DC)

#### C.2.1  SMOOTHING MODULE

The smoothing module is defined with an EMA operation (see equation 5), which performs several roles:

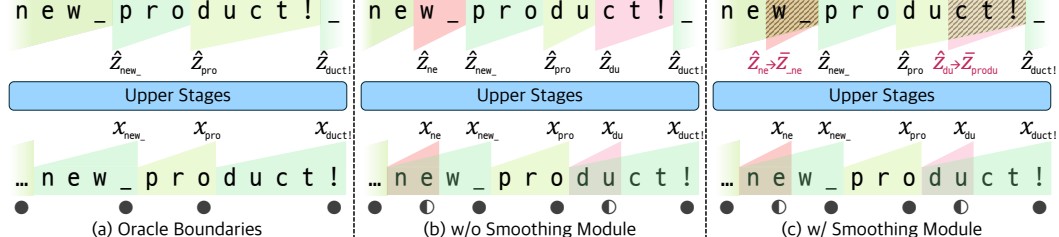

Figure 4: Comparison of decompression strategies on the example sequence `"...new product!"`. ● indicates a boundary with high confidence ($P_t = 1.0$) and ◐ indicates a boundary with low confidence ($P_t = 0.5$). As each letter in the example is unique, we use the letters in subscripts to denote expected semantics of chunks. (a) Optimal chunking with oracle boundaries identifying linguistically meaningful units. (b) Suboptimal chunking without a smoothing module. This creates misalignment during upsampling, causing information from incorrect contexts to propagate. (c) Improved decompression with a smoothing module, where low-confidence chunks are interpolated with weighted combinations of previous chunks, correcting the shaded regions. In panels (b) and (c), we interpret low-confidence boundaries cause the encoder network to embed broader contexts at subsequent positions. Specifically, the vectors at ⎵ and ! encode new⎵ and duct!, respectively (instead of w⎵ and ct!).

- **Differentiable boundary learning:** It transforms the discrete upsampling operation into a continuous one, enabling effective backpropagation through chunk boundaries during training without requiring stochastic exploration-based approaches (Jang et al., 2017).

- **Adaptive error correction:** Chunks with high confidence ($P_t \approx 1.0$) maintain discrete boundaries ($\bar{z}_t \approx z_t$), while chunks with low confidence ($P_t \approx 0.5$) are smoothed using information from previous chunks, creating a self-correcting mechanism.

- **Training stability:** By smoothly interpolating between discrete choices based on confidence scores, a smoothing module prevents the model from overfitting to suboptimal chunking patterns early in training.

Figure 4 illustrates this with the example `"...new product!"`. The word "product" can be morphologically decomposed into "pro-" and "-duct"[7]. Without the smoothing module (see Figure 4-(b)), suboptimal chunking (*e.g.,* `"du"` as shown with half-filled circles) creates alignment mismatches that disrupt information flow. With the smoothing module (see Figure 4-(c)), chunks with low confidence are smoothed with previous context, ensuring proper information propagation and enabling the model to learn optimal chunk boundaries through gradient descent.

### C.2.2 UPSAMPLER

Equations (6) to (9) are designed carefully with these following objectives:

- **Confidence scoring** equation 6: The coefficient $c$ quantifies the routing module's confidence in its boundary decisions. For positions marked as boundaries ($b_t = 1$), $c_t = p_t$ rewards high boundary probabilities. In contrast, for non-boundary positions ($b_t = 0$), $c_t = 1 - p_t$ penalizes false boundary predictions. This formulation encourages the model to produce boundary probabilities near 1.0 at true boundaries and near 0.0 elsewhere.

- **Gradient stabilization** equation 7: The Straight-Through Estimator (STE) (Bengio et al., 2013) is a well established technique from discrete representation learning (Van Den Oord et al., 2017; Jang et al., 2017) that rounds confidence scores to 1.0 in the forward pass while maintaining continuous gradients during backpropagation. While H-Net already demonstrates strong performance without STE, incorporating this technique provides an additional performance boost that empirically further stabilizes the optimization dynamics.

- **Causal expansion** equation 8: The upsampling operation repeats each compressed vector until the next boundary position, ensuring that each reconstructed position receives information from its most recent chunk. This maintains the sequential flow of information while expanding the compressed representation back to its original length.

- **Confidence-weighted decompression** equation 9: Multiplying upsampled vectors by their confidence scores incentivizes the routing module to make confident, accurate decisions.

---

[7]**pro-** – meaning *forward* or *forth*, **-duct** – from Latin *ducere*, meaning *to lead* or *to bring*

High-confidence boundaries create direct reward signals that encourage the model to sharpen its boundary predictions through gradient feedback.

### C.3 IMPROVED TECHNIQUES FOR HIERARCHICAL SEQUENCE MODELING

We introduce several techniques that improve the overall architecture. These may generally be considered techniques to improve *signal propagation* throughout the network, improving stability and learnability.

**Norm Balance.** Modern large language models employ pre-normalization architectures (Radford et al., 2019; Touvron et al., 2023a), departing from the post-normalization design of the original Transformer (Vaswani et al., 2017). Following established best practices, these models typically include a final normalization layer after all residual blocks. H-Net adopts this convention through *network normalization*, by placing an RMSNorm (Zhang & Sennrich, 2019) at the end of each network component ($\mathcal{E}^s$, $\mathcal{D}^s$, and $\mathcal{M}$).

This addition of a normalization layer addresses a critical challenge in hierarchical architectures. Pre-normalization allows residual stream magnitudes to grow unbounded through successive layers, with feature norms increasing monotonically. For H-Net, this poses a particular problem: the architecture leverages residual connections to preserve fine-grained information across stages. Without network normalization, outputs from deeper components (especially the many-layered main network) would dominate the residual signals from earlier encoder networks through imbalanced feature norms, neglecting the fine-grained details that are essential for decompression. The normalization layers restore balance between processed features and residual information, ensuring both contribute meaningfully to the final representation.

**Separation of Two Streams.** Encoder outputs ($\hat{x}$) serve dual purposes in our architecture: passing fine-grained information to corresponding decoders through residual connections, and providing compressed representations as inputs to subsequent stages. This dual functionality creates a design challenge, as these two roles may benefit from different representations. We consider three options to address this: (i) apply a projection to the residual connection only, (ii) apply a projection to the main network inputs only, (iii) and apply a projection to both pathways.

As indicated in equation equation 3, we adopt the first approach – adding a projection (Linear) only to the residual connection. This choice is motivated by the fundamental principle of designing deep learning models (He et al., 2016): maintaining intact gradient flow through the main computational path is crucial for effective training.

Empirically, we found that the third option underperforms despite additional parameters and computations, as the extra projections interfere with gradient propagation. The second option, while preserving residual gradients, disrupts the main network's gradient flow and had worse training dynamics. Our chosen design maintains unimpeded gradients from deeper stages while allowing the residual connection to adapt its contribution through the learned projection. This encourages the model to leverage the main network's computational depth while using residuals in a complementary role.

One additional detail is that this residual connection is initialized close to 0; earlier versions of H-Net found this to be an important detail, but it may be less important when combined with additional techniques such as LR modulation.

**Learning Rate Modulation** The hierarchical design of H-Net requires careful adjustment of learning rates across stages to ensure balanced training dynamics. Modern theory establishes that neural network hyperparameters should be scaled in predictable ways for optimal trainability (Yang & Hu, 2020). To provide a more systematic experimental results across different architectural configurations, we follow previous works and set learning rates to be proportionally to the (1) square root of batch size (Malladi et al., 2022; Merrill et al., 2025), and (2) inverse square root of hidden dimension (Vaswani et al., 2017; Yang & Hu, 2020). Concretely, without heavy manual tuning, we define $\lambda^s$ as follows:

$$\lambda^s = \sqrt{N^{\text{GPT}} \cdot \frac{\prod_{i=s}^{S} N^i}{\prod_{i=0}^{S} N^i} \cdot \frac{D^{\text{S}}}{D^s}}, \qquad N^S = 1.0 \tag{11}$$

where $N^{\text{GPT}}$ is the average number of bytes per token of training dataset, which is $4.6$ for the GPT-2 tokenizer on FineWeb-Edu.

With this modulation, the model achieves more stable training dynamics and improved convergence behavior across the entire hierarchy. In particular, we empirically find that since outer stages directly influence the chunk boundaries that inner stages depend on, the higher learning rates in the outer stages seem to accelerate learning the chunking mechanism. We note that such principles for optimizing signal propagation as neural network hyperparameters change is an active area of research, and our scaling factors are just heuristics that can likely be improved.

### C.4 AUTOREGRESSIVE TRAINING AND INFERENCE

In this section, we explain how H-Net preserves autoregressive properties throughout its hierarchical structure during both training and inference.

Every component of H-Net (*i.e.,* encoder-, decoder-, main- networks, and the dynamic chunking mechanism) is carefully designed to preserve autoregressive properties essential for language modeling.

**Training.** During training, H-Net employs standard causal masking across all sequence mixing layers. DC maintains causality by computing boundary probabilities based only on current and previous representations. Specifically, the boundary probability $p_t$ depends on $q_t$ and $k_t$ from the current and previous positions (equation equation 4), ensuring no information leakage from future tokens. The smoothing module similarly maintains causality through its recursive formulation (equation equation 5), where each output depends only on past compressed representations.

**Inference.** For inference, H-Net generates raw bytes (or whatever the outermost modality is) autoregressively with a modified procedure to handle its hierarchical structure.

Generation with a prompt proceeds as follows:

1. **Initial processing:** During prefill, we generate chunks via the encoders (as in training). For each component (i.e. the isotropic components, and the routing module and dechunking layer), we generate a state. Isotropic state (e.g. KV cache for Transformer layers, SSM state for Mamba-2 layers) is generated as usual.

2. **DC state and DC step:** As noted above, the DC modules have recursive formulations that maintain causality at train-time. These recursive formulations become autoregressive formulations at inference time.

   (a) **Routing Module:** In order to compute $p_t$, we need $k_{t-1}$ (see equation equation 4), so our state consists of the key value of the most recent token processed.

   (b) **Dechunking Layer:** In order to compute $\tilde{z}_t$, we need $P_t$ and $\tilde{z}_{t-1}$. Thus, the dechunking layer state should consist of the last $\tilde{z}$ value.

3. **Token Generation:**[8] To perform a model step, we do the following for a 1-stage hierarchy:

   (a) Pass the token through the encoder network,

   (b) Step the routing module to determine whether the token needs to be processed by the main network,

   (c) Step the main network if necessary, in which case we also need to step the dechunking layer.

   (d) Use the result of the dechunking layer to step the decoder network.

A consequence of this inference formulation is that, at inference time, H-Net decides individually for each token how much compute to use when processing it. Therefore, H-Net can allocate more or less compute to different tokens as it deems necessary. A particular connection is that inference resembles speculative decoding (Leviathan et al., 2023; Chen et al., 2023), which also involves a small network (the *draft model*) stepping on every token, and a larger network (the *verification model*) only stepping on contiguous chunks of every few tokens.

---

[8]Here, we use *token* in the autoregressive generation sense, referring to one time step, not in the literal BPE token sense.

# D  ADDITIONAL EXPERIMENTAL DETAILS

## D.1  FLOPS COMPUTATION

We largely follow Hoffmann et al. (2022) with two marginally updated computations: (1) add computations for Mamba-2 (Dao & Gu, 2024), and (2) modify computations in MLP blocks as we use the recent Transformer++ architecture. Assuming that all query, key, and value share the same num_heads and head_dim, we calculate the forward pass FLOPs as follows:

- **Embeddings:** $2 \times \text{seq\_len} \times \text{vocab\_size} \times \text{d\_model}$
- **Attention:**
    - $QKV$ **projections:** $2 \times 3 \times \text{seq\_len} \times \text{d\_model} \times (\text{num\_heads} \times \text{head\_dim})$
    - **Attention Logit Calculation:** $2 \times \text{seq\_len} \times \text{seq\_len} \times (\text{num\_heads} \times \text{head\_dim})$
    - **Attention Score Softmax:** $3 \times \text{num\_heads} \times \text{seq\_len} \times \text{seq\_len}$
    - **Score @ Query:** $2 \times \text{seq\_len} \times \text{seq\_len} \times (\text{num\_heads} \times \text{head\_dim})$
    - **Output projection:** $2 \times \text{seq\_len} \times (\text{num\_heads} \times \text{head\_dim}) \times \text{d\_model}$
- **Mamba-2:**
    - $XZ$ **projections:** $2 \times \text{seq\_len} \times \text{d\_model} \times (2 \times \text{expand} \times \text{d\_model})$
    - $BC\Delta t$ **projections:** $2 \times \text{seq\_len} \times \text{d\_model} \times (2 \times \text{d\_state} + \text{num\_heads})$
    - **SSD:** $2 \times 3 \times \text{seq\_len} \times (\text{expand} \times \text{d\_model}) \times \text{d\_state}$
    - **Depthwise Convolution:** $2 \times \text{seq\_len} \times \text{d\_model} \times \text{window\_size}$
    - **Gating:** $5 \times \text{seq\_len} \times \text{d\_model}$
    - **Output projection:** $2 \times \text{seq\_len} \times \text{d\_model} \times \text{d\_model}$
- **Gated MLP:**
    - **In, Gate, Out projections:** $2 \times \text{seq\_len} \times (3 \times \text{d\_model} \times \text{ffw\_size})$
    - **Gating:** $5 \times \text{seq\_len} \times \text{d\_model}$
- **Logit Prediction Head:** $2 \times \text{seq\_len} \times \text{vocab\_size} \times \text{d\_model}$

We assume the backward pass consumes twice the FLOPs of the forward pass.

## D.2  ROBUSTNESS SCORE

We introduce a metric called the *robustness score* to measure the robustness of a model's performance to textual perturbations, defined for Hellaswag as follows:

$$\text{robustness score} := 100 \cdot \frac{\text{perturbed accuracy} - 0.25}{\max(\text{unperturbed accuracy} - 0.25, 0)}.$$

This score measures the percentage of original (unperturbed) performance that is captured by the model in the perturbed setting. We subtract by $0.25$ as HellaSwag is multiple choice with 4 options, thus a model that scores $0.25$ in the perturbed setting should be considered to have lost all of its original capability.

## D.3  EXPERIMENTAL SETUP FOR CHINESE AND CODE.

In Section 3.2, we analyzed the performance of H-Net (2-stage) against Transformer and H-Net (space) on Chinese and on code, finding superior scaling for H-Net (2-stage) versus the other architectures. Here, we describe additional details from the experiment.

On Chinese and code, we use 46B token subset from FineWeb-Edu-Chinese-V2.1 (Yu et al., 2025) and Github subset from Pile (Gao et al., 2020) to train three models at the 1.3B GPT-3 *XL* scale: H-Net (2-stage), H-Net (space), and Transformer. We maintain the same bytes per gradient step (256 batch size with 8192 `utf-8` encoded bytes per example) as the main text experiments.

For model architecture, we primarily matched the settings from the GPT-3 *XL*, including $d_{\text{model}}$ and encoder/decoder architecture for H-Net models. However, we adjusted the number of layers in the main network of each model to account for slightly different compression ratios. Specifically, the Chinese-language models used a slightly higher total training flops target than the original language models, while the code models used a lower flops target. Full architecture details and results are also in Table 5.

Table 5: **Architecture details and model benchmarks for Chinese and code models.** BPIC (defined in Table 2) denotes the compression between the main network and outermost stage (bytes). Each H-Net used (3,3)-DC, targeting an inner downsampling ratio of 9. However, the resulting BPIC was significantly different, indicating that code is much easier to compress than Chinese. In terms of results, H-Net (2-stage) performs better than both H-Net (space) and BPE Transformer on Chinese, which is reflected in the downstreams. On the other hand, H-Net (2-stage) achieves similar performance to H-Net (space) on code, and both H-Net models perform significantly better than Transformer.

| MODEL | CHINESE | | | | CODE | | |
|---|---|---|---|---|---|---|---|
| | BPIC | MAIN ARCH. | VAL. BPB ↓ | XW-ZH. ACC. ↑ | BPIC | MAIN ARCH. | VAL. BPB ↓ |
| Transformer | 3.62 | T15 | 0.7404 | 0.599 | 3.58 | T13 | 0.3376 |
| H-Net (space) | 3.38 | T19 | 0.7478 | 0.629 | 7.97 | T40 | 0.3163 |
| H-Net (2-stage) | 5.81 | T30 | **0.7032** | **0.663** | 7.23 | T28 | **0.3161** |

For the H-Net (2-stage), we use the same target downsampling ratio ($N^0 = N^1 = 3$) as the main experiments. Unlike BPE or spacelike-based tokenization, whose downsampling ratios can vary widely by dataset, H-Net allows for using similar compute budgets without much adjustment. For H-Net (space), we use the same definition of spacelike as the original SpaceByte paper (Slagle, 2024), and for BPE, we use the Llama3 tokenizer (Grattafiori et al., 2024), as the GPT2 tokenizer attains very poor downsampling ratios on both datasets. Despite this change, both H-Net (space) and Transformer (BPE) still have highly varied downsampling ratios between ordinary (primarily English) language, Chinese, and code. On the other hand, H-Net can adhere to a target ratio regardless of dataset, chunking into concepts at appropriate ratios.

| MODEL / ARCHITECTURE | PARAMS. | FINAL PPL. ↓ |
|---|---|---|
| Transformer ( T9 ) | 29M | 2.769 |
| Mamba-2 ( M10 ) | 33M | 2.757 |
| H-Net ( M3T1 + T15 + M4 ) | 64M | 2.705 |
| H-Net ( M3T1 + M15 + M4 ) | 66M | 2.697 |

Table 6: **Model details and final performance on HG38.** We trained two isotropic models and two H-Net models, varying the main network architecture (Transformer or Mamba-2). Each H-Net model outperforms the corresponding isotropic model. We empirically find that the $\mathcal{E}^0$ = M3T1 encoder architecture slightly outperforms a pure Mamba-2 encoder $\mathcal{E}^0$ = M4 (Section E.3.7).

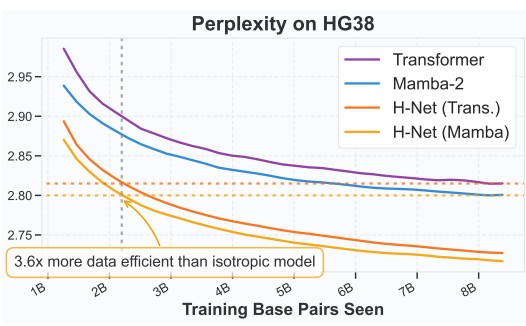

Figure 5: **Scaling performance on HG38** during the stable phase of training. Each H-Net model achieves the same pre-decay perplexity of the corresponding isotropic model with approximately $3.6\times$ less data.

Table 7: **Distilling Llama 3.2 3B to a byte level model.** Average acc indicates average of the benchmarks measured in Table 2. H-Net loses performance across the board compared to the teacher, which is expected because we cannot quite replicate the exact behavior of the original model due to non-causality of BPE tokens. However, it is still much stronger than an H-Net trained from scratch on this small amount of data (189B bytes).

| MODEL | LMB. ACC ↑ | HELLA. ACC_n ↑ | PIQA ACC ↑ | ARC-E ACC ↑ | ARC-C ACC_n ↑ | WINO. ACC ↑ | OPEN. ACC_n ↑ | AVERAGE ACC ↑ | MMLU (5-SHOT) ACC ↑ |
|---|---|---|---|---|---|---|---|---|---|
| Llama 3.2 3B (base) | 0.701 | 0.737 | 0.768 | 0.745 | 0.460 | 0.688 | 0.428 | 0.647 | 0.561 |
| Distilled H-Net (1-stage) | 0.634 | 0.702 | 0.761 | 0.721 | 0.433 | 0.665 | 0.414 | 0.617 | 0.519 |

## E ADDITIONAL EXPERIMENTS

### E.1 DNA (HUMAN GENOME) EXPERIMENTS

DNA is a setting that presents both a unique promise and challenge for hierarchical modeling. For one, handcrafted tokens do not work well on DNA, due to the lack of segmentation cues. Additionally, the same sequence of base pairs may serve different functions (*e.g.,* depending on whether or not the pair is inside a gene or not). Consequently, a naive BPE-based approach may not work either. On the other hand, DNA *can* exhibit higher resolution structure (*e.g.,* codons, various regulatory elements), suggesting that there is room for principled hierarchical modeling. Indeed, state-of-the-art DNA models (Brixi et al., 2025) operate directly on base pairs (A, C, G, T) with implicit hierarchical structure.

Thus, we evaluated four models on DNA: two isotropic models (pure Transformer and pure Mamba-2) operating at the base-pair level, and two corresponding H-Net (1-stage) with Transformer and Mamba-2 as the main network. Each model was trained on the HG38 dataset with a learning rate of $5 \cdot 10^{-3}$ for modules at the base-pair resolution. For the H-Net models, we used a downsampling ratio of $N^0 = 3$. All models were trained with a $d_{\text{model}}$ of $512$, which was used for all isotropic modules of H-Net (including the main network).

Previous work has shown that SSMs show improved DNA modeling ability compared to Transformers (Gu & Dao, 2024), and we find that this effect is preserved when examining Transformers vs. Mamba-2 as the main network (see Table 6). This finding suggests that existing layer selection principles can be applied when deciding main network architecture. In fact, by directly comparing the perplexity curves during the stable phase of training (Figure 5), we find that H-Net models can achieve similar performance to isotropic models with $3.6\times$ less data, a finding that holds for both choices of main network architecture.

### E.2 DISTILLING TOKEN-LEVEL MODELS TO BYTE-LEVEL

The role of the outer stages in H-Net is analogous to that of the tokenizer, embedding module, and LM head in a traditional BPE-based language model; together, these modules interconvert between raw text and an embedding space that the main model backbone can process. Given this similarity,

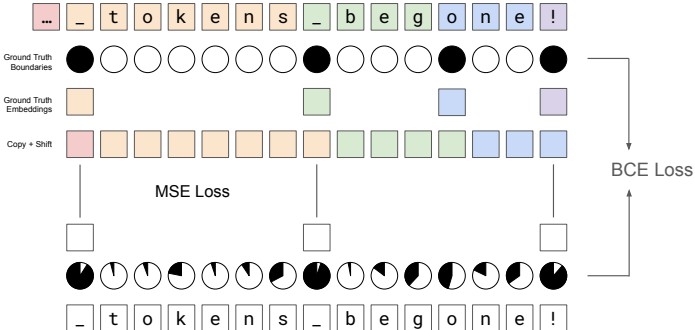

Figure 6: **Auxiliary loss strategy for training the encoder** of a H-Net with pretrained main stage. In order to mimic the behavior of the tokenizer + embedding layer of a pretrained language model, we add supervision to both the routing module boundary probabilities and to the hidden states that we pass through to the main network. These losses encourage the encoder to tokenize once at the start of every token, while also passing the correct embedding into the main network near the start of the token, thus making maximal use of the next-token prediction ability.

we investigated whether it would be possible to convert a BPE-tokenized model directly into a byte level H-Net. To do this, we trained a 1-stage H-Net with frozen main network initialized from the backbone of Llama 3.2 3B (base). Our H-Net uses 4 Mamba-2 layers without MLPs for both the encoder and decoder with a hidden dimension of 1536. Because the Llama model has a hidden dimension of 3072, we add MLP adapters with hidden dimension 8192 after chunking and right before dechunking (i.e. right before and after feeding into the main stage). We train the model for 90000 gradient steps with sequence length 8192 and batch size 256, for a total of 189B bytes.

**Aligning the encoder.** The primary difficulty in converting a tokenized model into a byte-level one is that the encoder and DC must produce chunks that the tokenized model can produce useful output with. Thus, our training (besides using the standard next-byte prediction loss), adds the following losses (see Figure 6 for a visual).

1. A binary cross-entropy boundary-prediction loss (with equal weight as the main loss) that operates on the routing module probabilities and targets the router to pass *the start of every real token* through the main network.

2. A hidden state matching loss that matches the post-adapter hidden state with the "correct" hidden state. Here, if $\hat{z}_k$ is the hidden representation that was passed into the main network at (byte) position $t$, we try to match $z_k$ with the embedding of the token that the $t$th byte was part of, *except* when the $t$h byte is the first byte of its token, in which case we match the $z_t$ with the *previous* token's embedding. Embedding matching is done with an L2 loss with a weight of 0.02.

In the ideal case where both losses are zero, the router sends exactly the first byte of each token through to the main network with the right embedding. The main network would thus see exactly the representation it would see with a tokenizer + embedding setup. In practice, sending both losses to zero is literally impossible, as we discuss below. However, we still find that the boundary-prediction loss is crucial for learning a good matching, while the embedding-matching loss is helpful in speeding up training but not necessary. In fact, increasing the loss weight on the embedding-matching loss too much can harm the language-modeling loss.

**Tokenization bias.** We are not able to send all auxiliary losses to zero because online prediction of BPE boundaries is an impossible task. Phan et al. (2025) coined the term "tokenization bias" to represent the fact that the tokenization process implicitly contains next-byte information. For example, the Llama 3 tokenizer tokenizes the strings ␣distill and ␣distinct into [␣dist, ill] and [␣distinct]. Prior use of this term has been to suggest that if an autoregressive

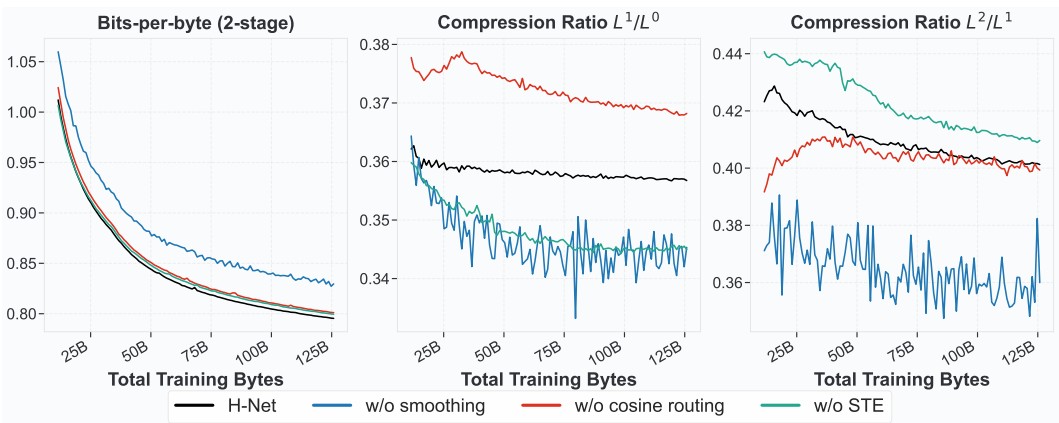

Figure 7: **Ablation study on key H-Net components** showing validation BPB (left) and compression ratios for the first stage $L^1/L^0$ (center) and second stage $L^2/L^1$ (right) during training. Using H-Net (2-stage), we evaluate the impact of removing three components: the smoothing module (w/o smoothing), the similarity-based routing module (w/o cosine routing), and Straight-Through Estimator (w/o STE).

language model is prompted with `_dist`, the nature of its training will be that it will never complete with `inct` (this is in fact a flaw of all tokenization-based models).

For us, however, tokenization bias implies that we cannot determine whether or not the `i` in `_disti` is the start of a new word until *after* seeing the next character. In fact, the problem can be even worse–consider `_applicable` (becomes [`_app`, `licable`]) and `_applicant` (becomes [`_applicant`]): Determining whether `l` is the start of a token requires knowing the next two bytes as well.

While the H-Net does use the main network, it is not able to exactly match the behavior of the original tokenized model. Instead, it is finding slightly different representations of tokens to use in the main stage. Recent work has shown that tokenized language models can process tokenization sequences distinct from the "canonical" greedy tokenization (Vieira et al., 2024), so it is possible our H-Net found another alternate representation that the pretrained model could process.

**Remark.** One might ask if our distilled model has simply learned to tokenize on spaces (since spaces are always the start of a new token). It has not. Simply tokenizing on spaces would yield a sub-95% boundary prediction accuracy; however, our distilled model gets boundary prediction accuracy above 99.5%. This suggests that the resulting H-Net is able to recognize some, but not all, subword boundaries.

**Results.** The results from our distillation procedure are shown in Table 7. H-Net is able to approximately match performance across almost all benchmarks; in general, H-Net is not able to replicate the behavior of the tokenized model exactly, so it is not unexpected that the benchmarks are slightly worse. Byte-Latent Transformer (Pagnoni et al., 2024, Table 5) performs a similar experiment, and they see a greater gap among most benchmarks (particularly PiQA, Arc-Easy, and Arc-Challenge) despite using a much larger amount of data (220B *tokens* versus 189B *bytes*); it is possible that this performance difference is due to the fact that a BLT module cannot be supervised to align boundaries the way that end-to-end DC can.

### E.3 ABLATION STUDIES

In this section, we provide comprehensive ablations that study individual architectural components and design choices. For ablation studies, we employ H-Net at *Large* scale following the configurations in Table 1, training on 36B tokens randomly sampled from FineWeb-Edu.

#### E.3.1 IMPORTANCE OF COMPONENTS IN H-NET

Figure 7 illustrates the impact of each architectural component on both model performance and compression ratio ($L^{s+1}/L^s$) stability during training. We conduct three targeted ablations: (i) using direct upsampling $\tilde{z}_t = \hat{z}_t$ by removing the smoothing module (*w/o smoothing*), (ii) replacing

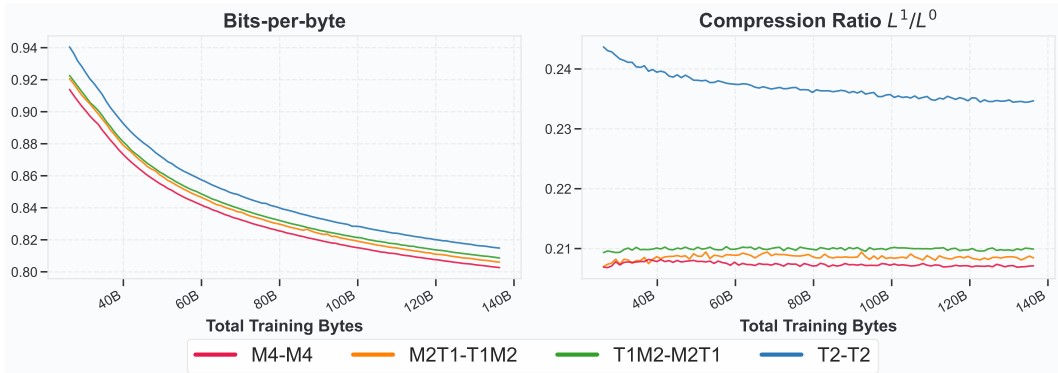

Figure 8: **Encoder-decoder architecture ablation using raw byte inputs.** Validation BPB (left) and compression ratio $L^1/L^0$ (right) for H-Net (1-stage) throughout training. We evaluate four encoder-decoder ($\mathcal{E}^0 - \mathcal{D}^0$) configurations: M4-M4 , M2T1-T1M2 and T1M2-M2T1, and T2-T2, where M denotes Mamba layers and T denotes Transformer layers.

the routing module that is based on scaled cosine similarity, with direct probability prediction from individual inputs (*w/o cosine routing*), and (iii) skipping the straight-through estimator in equation equation 9 (w/o STE).

The smoothing module proves essential for stable training dynamics. Without this module, compression ratios fluctuate severely throughout training, preventing the model from learning consistent chunking boundaries. This instability directly manifests as substantial performance degradation, confirming that smooth gradient flow through the decompression process is crucial for effective end-to-end learning. While less critical than the smoothing module, both the similarity-based routing module and STE operation exhibit importance in training stability and final performance. These components help maintain consistent compression ratios and lead to more interpretable chunking patterns. The similarity-based approach particularly enables the model to identify natural linguistic boundaries (*e.g.,* whitespaces, subwords) by comparing adjacent representations rather than making isolated predictions.

### E.3.2 ENCODER & DECODER LAYER SELECTION

The composition of sequence mixing layers in H-Net's encoders and decoders substantially influences both compression efficiency and modeling capability. We systematically evaluate different architectural combinations using H-Net (1-stage) while fixing all other configurations in Table 1 the same. Four distinct encoder-decoder ($\mathcal{E}^0$-$\mathcal{D}^0$) pairings are tested: M4-M4, M2T1-T1M2, T1M2-M2T1, and T2-T2, where M denotes a Mamba-2 layer and T denotes a Transformer layer. These combinations are chosen by keeping the symmetry and replacing each Transformer layer with two Mamba-2 layers, as they contain equivalent parameter counts — $12D^2$ for Transformer ($4D^2$ for the attention mechanism and $8D^2$ for an MLP) vs. $\approx 6D^2$ per Mamba-2 layer (no MLP).

Figure 8 and Figure 9 demonstrate that Mamba layers are essential for effective byte-level sequence processing. For both H-Net and SpaceByte++, **the pure Transformer configuration (T2-T2) exhibits by far the worst performance despite using more FLOPs** (it also down-compresses sequences poorly compared to other configurations, thus using more compute in the main network). This configuration struggles to compress byte sequences effectively, resulting in both computational waste and degraded modeling performance. Performance improves monotonically with increased Mamba layer allocation, achieving optimal results with the highest compression efficiency in the pure Mamba configuration (M4-M4). These findings align with recent research demonstrating SSMs' advantages over Transformers for fine-grained sequence modeling (Goel et al., 2022; Schiff et al., 2024), as corroborated by MambaByte's superior performance over LlamaByte in Figure 2.

A natural question arises: does the importance of Mamba layers (i) stem specifically from **processing fine-grained byte inputs**, or (ii) because **they are better for compressing information into the next stage, even at coarser input resolutions**? To investigate these hypotheses, we train a 1-stage H-Net on top of *BPE-tokenized* inputs processed by the GPT-2 tokenizer. We then evaluate six different encoder-decoder combinations.

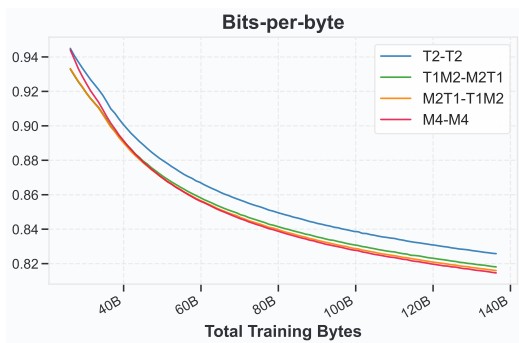 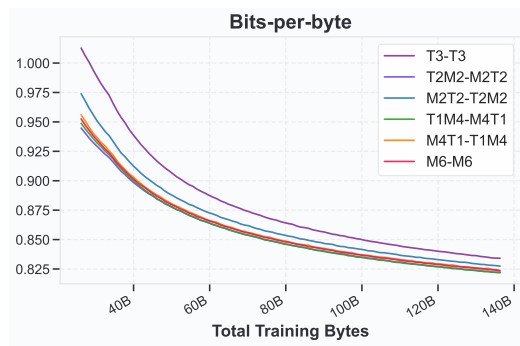

Figure 9: **SpaceByte++ encoder-decoder architecture ablation using raw byte inputs.** We evaluate four encoder-decoder ($\mathcal{E}^0 - \mathcal{D}^0$) configurations: M4-M4 , M2T1-T1M2 and T1M2-M2T1, and T2-T2, where M denotes Mamba layers and T denotes Transformer layers.

Figure 10: **Encoder-decoder architecture ablation using BPE-tokenized inputs.** Assuming that GPT-2 tokenizer serves as the outermost encoder-decoder (*i.e.,* $\mathcal{E}^0$-$\mathcal{D}^0$), we evaluate six $\mathcal{E}^1$-$\mathcal{D}^1$ combinations: M6-M6, M4T1-T1M4, T1M4-M4T1, M2T2-T2M2, T2M2-M2T2, and T3-T3.

- If hypothesis (i) holds, then we would expect different combinations of Mamba/Transformer layers in the encoder/decoder to have similar performance, since it is known that they have similar performance on standard tokenized language modeling.

- If hypothesis (ii) holds, then we would expect that encoders/decoders using some Mamba layers to be better than pure Transformer layers.

As demonstrated in Figure 10, Mamba layers prove significantly important even when processing BPE tokens rather than raw bytes, providing evidence for the second hypothesis.

We hypothesize that this consistent advantage across input granularities stems from fundamental architectural differences between SSMs and Transformers. While Transformers naturally store complete key-value caches for all positions, SSMs are designed to compress information into fixed-size states. This compression-oriented architecture aligns naturally with our chunking mechanism, which requires aggregating multiple input vectors into consolidated representations. The inherent compression capability of Mamba layers makes them particularly well-suited for the encoder and decoder roles in our hierarchical architecture (Gu, 2025). Based on these findings, we employ Mamba layers throughout all encoders and decoders in our final H-Net configuration, as detailed in Table 1.

These findings transfer to more general hierarchical structures (such as a 2-stage H-Net at the byte level), in which case the outermost encoder and decoder layers ($\mathcal{E}^0$ and $\mathcal{D}^0$) serve a similar role as the GPT-2 tokenizer and the inner layers ($\mathcal{E}^1$ and $\mathcal{D}^1$) would share similar findings of benefiting from using Mamba layers.

### E.3.3 VISUALIZATION OF TOKENIZED POSITIONS

In Figure 11, we provide visualizations of the boundaries dynamically drawn by H-Net (1-stage) and H-Net (2-stage). The visualization offers several insights about how the model decides boundaries.

- *Single-stage behavior:* H-Net (1-stage) predominantly places boundaries at whitespace characters, closely mirroring the delimiters used by SpaceByte. This indicates that H-Net learns that word boundaries represent natural semantic units in text. This convergence to spacelike boundaries, discovered purely through end-to-end training, conversely validates SpaceByte's strong empirical performance.

- *Hierarchical chunking patterns:* The first stage of H-Net (2-stage) combines spacelike boundaries with first few characters of each word. This strategy helps the model because once the initial positions of a word are identified, the remaining characters become highly predictable.

- *Content-aware chunking:* One might question if H-Net's chunking decisions follow static rules, such as drawing boundaries only at certain fixed bytes (*e.g.,* whitespace). However,

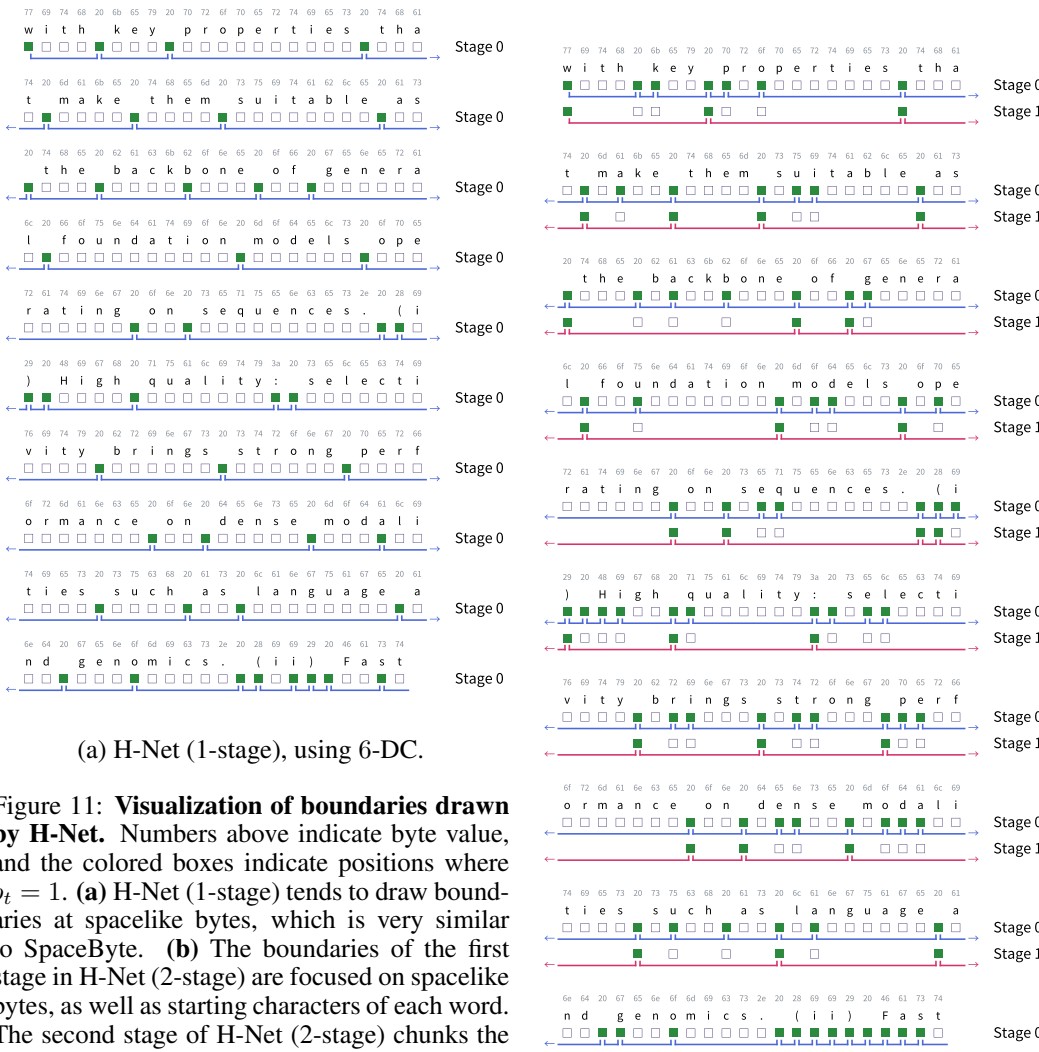

(a) H-Net (1-stage), using 6-DC.

Figure 11: **Visualization of boundaries drawn by H-Net.** Numbers above indicate byte value, and the colored boxes indicate positions where $b_t = 1$. **(a)** H-Net (1-stage) tends to draw boundaries at spacelike bytes, which is very similar to SpaceByte. **(b)** The boundaries of the first stage in H-Net (2-stage) are focused on spacelike bytes, as well as starting characters of each word. The second stage of H-Net (2-stage) chunks the text into more meaningful units, such as words or numberings (*i.e.,* `(ii)`). We can also observe that it often chunks multiple words that form one semantic group; for example, `the backbone` and `such as`.

(b) H-Net (2-stage), using (3,3)-DC.

as shown in the figure, H-Net often merges multiple words and spacelike characters based on content (examples include `the backbone`, `such as`, and `(ii)`).

- *Perturbation behavior:* Figure 12 shows the same example with textual perturbations such as removing whitespaces, which more prominently demonstrates that boundaries drawn by H-Net are based on content and context. In particular, it often still chunks in between semantic words even if the space is removed.

### E.3.4 HYBRID ARCHITECTURES FOR THE MAIN NETWORK

We also aimed to understand the role of architecture selection in the main network. To this end, we compared H-Net (2-stage) with an identical model where we replaced the Transformer stack with a hybrid model containing both 20 Mamba-2 and 7 Transformer layers interleaved in a 3:1 ratio. Hybrid architectures have shown promise in isotropic (BPE) models (Waleffe et al., 2024), and similarly perform better for our choice of main network (Figure 13).

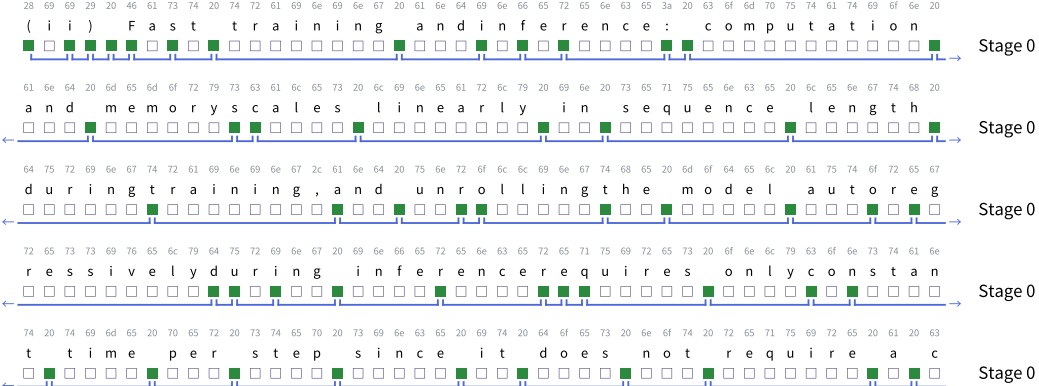

Figure 12: Visualization of boundary positions dynamically drawn by H-Net (1-stage). The given text is perturbed that some whitespaces are missing. H-Net detects word boundaries even if they are not explicitly separated by whitespaces.

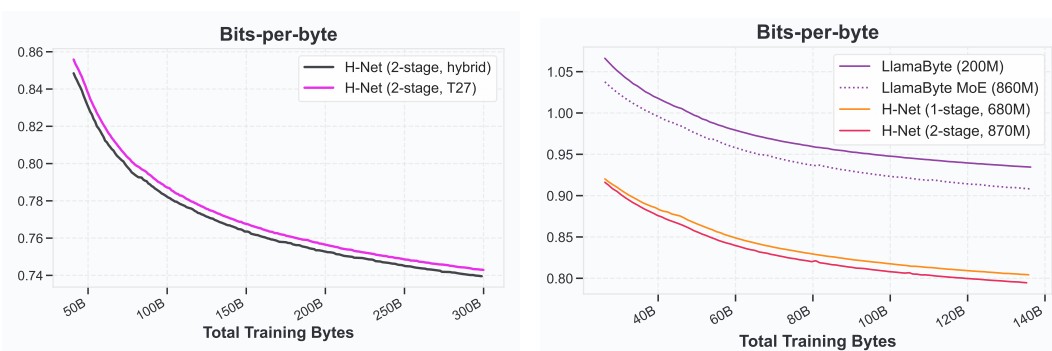

Figure 13: **Hybrid main network.** Bits-per-byte during the stable phase of training, for H-Net (2-stage) with Transformer main stage and with hybrid main stage. The hybrid main stage scales better, similar to findings for standard token-based language models. This finding suggests that design principles for isotropic (tokenized) models can carry over to choices of the main network.

Figure 14: **Comparison to Mixtures-of-Experts.** Bits-per-byte comparison of H-Net (both 1-stage and 2-stage) to LlamaByte-MoE, which is a FLOPs-matched MoE model that uses a similar number of parameters as H-Net (2-stage). Both H-Nets perform much better than LlamaByte-MoE, implying that H-Net's capabilities do not just come from sparsity.

### E.3.5 COMPARISON TO MIXTURE-OF-EXPERTS

H-Net can be viewed as a form of dynamic sparsity similar to Mixture-of-Experts (MoEs), in that they are able to improve performance by using more parameters, all while keeping the total FLOPs budget constant. We were interested in understanding whether or not its performance benefits were simply due to increasing sparsity. We compare against a sparsified version of LlamaByte (byte-level isotropic Transformer model) at the *Large* scale with a standard Mixture-of-Experts recipe (Fedus et al., 2022) and similar parameter count as ours (Figure 14). While sparsity does improve LlamaByte performance, it is still far worse than either FLOPs-matched H-Net (1-stage) or H-Net (2-stage), even with similar parameter count. We interpret this result as: H-Net not only achieves sparsity, but does so in a more *semantically meaningful manner*, which allows for better scaling than even generic sparse methods.

### E.3.6 DIFFERENT DOWNSAMPLING METHODS IN THE CHUNKING LAYER

Given the dynamically determined boundaries from the boundary predictor, we explore various compression strategies in the chunking layer. We compare the default Downsample operation of H-Net

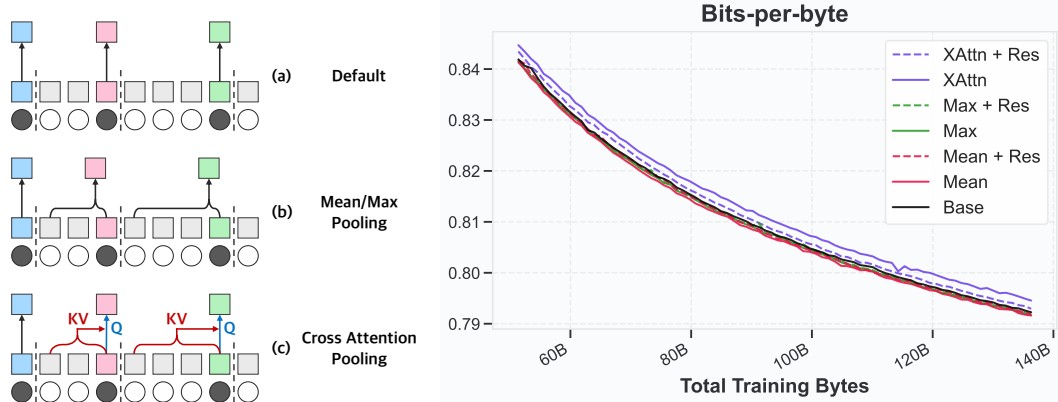

Figure 15: **Compression Methods in chunking layer.** Default: H-Net's Downsample operation **(left-a)**. Max/Mean: Channel-wise max and mean pooling within boundaries **(left-b)**. XAttn: Cross-attention pooling within boundaries **(left-c)**. `+Res`: Adds boundary vector residuals to compressed outputs.

| MODEL | ARCHITECTURE | PARAMS. | FINAL PPL. ↓ |
|-------|-------------|---------|--------------|
| H-Net | M3T1 + T15 + M4 | 64M | 2.705 |
| H-Net | M3T1 + M15 + M4 | 66M | 2.697 |
| H-Net | M4 + T15 + M4 | 62M | 2.722 |
| H-Net | M4 + M15 + M4 | 64M | 2.706 |
| H-Net | M4 + T1M13T1 + M4 | 64M | 2.706 |

Table 8: **Encoder architecture ablations on HG38.** Switching the encoder architecture from M3T1 to M4 leads to worse performance across the board, though the results are still better than isotropic models (Table 6). Transformers in the encoder network do not appear to be helpful for text (Figure 8), suggesting that this finding may be modality-specific.

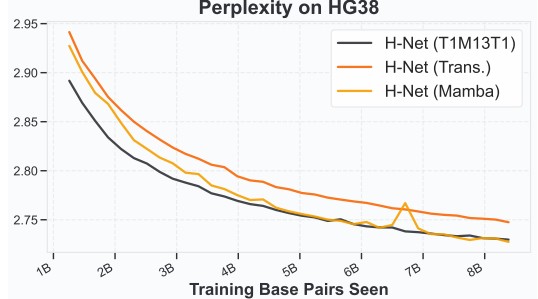

Figure 16: **Mamba-2-only encoder loss curves** during the stable phase of training. The pure Mamba-2 model is more unstable with a loss spike. Adding Transformer layers to the main network near the DC modules can alleviate instabilities. H-Net (1-stage, principled) corresponds to the T1M13T1 main network architecture.

(see Section 2.2.1) against three alternatives (see Figure 15-left): channel-wise max/mean pooling and cross-attention, all applied to vectors within the same boundary. Despite its simple design, the default compression in H-Net performs on-par with the other variants as demonstrated in Figure 15-right. This shows that the sequence mixing layers in encoder are trained to implicitly compress necessary context into vectors at boundaries, without explicit compression mechanisms such as pooling or cross-attention.

### E.3.7 DNA ARCHITECTURE ABLATIONS

As shown in Figure 5, H-Net (1-stage) with an M3T1 encoder achieves $3.6\times$ the data efficiency of an isotropic architecture. As mentioned in the caption of Table 6, we found that an M3T1 encoder outperformed a pure Mamba-2 M4 encoder, which is demonstrated in Table 8. The results in Figure 8 show that putting a Transformer in the encoder network does not appear to be helpful for text. Thus, it is possible the Transformer being useful is a DNA-specific result.

Interestingly, the loss curve for the M4 encoder with a pure Mamba-2 main network was more unstable. We then also tried replacing the M15 in the main network with a T1M13T1 architecture, inspired by the finding that Transformer layers are good for dealing directly with compressed input (see Figure 10). The new, principled main network architecture improved stability greatly as shown in Figure 16.

## F  LLM Usage for Paper Writing

LLMs were used only to detect typos and grammatical errors.

## G  Acknowledgements

We thank Nimit Sohoni, Eli Pugh, Justin Liu, Karan Goel, Arjun Desai, and Brandon Yang for feedback and support throughout the project. We thank Tri Dao for feedback on the ideas and earlier drafts. We thank Aakash Lahoti, Aviv Bick, Kevin Li, and Isaac Liao for feedback and discussions.

