# OpenReview forum: "Dynamic Chunking for End-to-End Hierarchical Sequence Modeling"
_ICLR.cc/2026/Conference — ICLR 2026 Poster_

### Official Review · Reviewer_7WMM · 2025-10-25

**Soundness:** 3
**Presentation:** 2
**Contribution:** 3
**Rating:** 6
**Confidence:** 3

**Summary:**

The authors propose a hierarchical model (H-net) that can operate on raw byte sequences. The model is learned end-to-end. First, the sequence is processed by a shallow Mamba-based encoder, then it is chunked and downsampled to reduce its length. Then, the main network, which can have any architecture —typically a Transformer or, recursively, an H-net for multiple hierarchy levels. Then, a smoothing module is applied to provide gradients for the decision points, the sequence is upsampled by repeating the main network's output the correct number of times, and a Mamba-based decoder network outputs the full-length sequence. Additional regularization is used to achieve the desired target compression ratio.

The model consistently outperforms its competitors, including Transformers, SpaceByte, and MambaByte

**Strengths:**

- Interesting, fully differentiable method
- Improved scaling
- Improved robustness

**Weaknesses:**

- Complexity
- Clarity: Some baselines are not described in enough detail. For example, what exactly is H-Net (Space) or H-Net (pool)? Some details are described in the appendix, but are very vague. Line 353 claims that the main network also uses Mamba layers, but this is never described. Are all layers Mamba layers? Only some?
- The only BPE model is a Transformer; however, the H-net uses Mamba layers. It would be nice to have a Mamba-based BPE model to see if the dynamic chunking or the Mamba is a bigger win.

**Questions:**

- In line 206, the authors say "where chunking layer ... ", however the chunking layer is defined afterward, which is a bit confusing
- In line 355, the authors say "As discussed Section, ... comprise mainly Mamba-2 layers.". However, Mamba was never mentioned before in the paper.
- It would be nice to describe what the scales are in the text. Now one has to look at the figure descriptions to be able to figure it out.
- In line 1455, the authors say "Multiplying upsampled vectors by their confidence scores incentivizes the routing module to make confident, accurate decisions.". However, they are multiplied by one, and the difference is only in the backward. Why does this trick help?

---

> ### Author Response · Authors · 2025-11-23
> **Rebuttal to Reviewer 7WMM**
>
> We sincerely thank the reviewer for their thoughtful assessment of our work. We appreciate the recognition of our fully differentiable hierarchical design, as well as the positive feedback on the improved scaling behavior and robustness of H-Nets.
>
> We now address the reviewer’s main concerns:
>
> > **Complexity**
>
> We acknowledge that our model is more complex than standard language models, which typically consist of a simple stack of Transformer or Mamba layers. However, we would like to note that
> **There are only two major differences on top of prior work** (the cosine-similarity routing module and the EMA-based smoothing module) each of which is technically motivated, and
> While these additions introduce some complexity, these changes together enabled new capabilities: the first truly end-to-end tokenizer-free model competitive with standard language models.
>
> We look forward to downstream work that can continue improving and simplifying H-Nets.
>
> > **Clarity and organization (appendix dependence, model variants, and notation).**
>
> We thank the reviewer for these detailed comments on clarity and organization. These suggestions should improve the readability of the paper. In our revised version, we added these definitions and references (e.g., H-Net (space)/(pool), Mamba usage, and scale descriptions) into the main text. Below, we note each specific point.
> - **What are H-Net (space) and H-Net (pool)?**
> H-Net (space) and H-Net (pool) share the same architecture design and training strategies as H-Net (1-stage); the only difference lies in the chunking mechanism. Concretely, aside from the routing module, the models are the exact sample.
> H-Net (space) uses SpaceByte’s delimiters for chunking, whereas H-Net (pool) uses fixed-interval chunking. We will explicitly define these variants in the main text.
> - **Main network layers**
> For the main language modeling experiments, the main network does not use Mamba layers (see Table 4). Mamba layers are only used in the main network for the DNA and hybrid model experiments. We will clarify this in the text to avoid future confusion.
> - **The chunking layer is mentioned before being defined (line 206).**
>  We agree this is indeed confusing. In the revision, we will introduce and define the chunking layer before using the term in the main text.
> - **Mamba is referenced without prior mention (line 355).**
> Thank you for catching this. The pointer should refer to Sec. C.1 in the appendix, where Mamba layers are introduced, rather than to Sec. 2. We will correct the reference and expand the main body’s methodological description in a camera ready version, given a potential extra page.
> - **Description of scales in the text.**
> In our revised text, we added parameter counts (Line 359) to the main text to clarify the sizes of the L and XL scales. We are happy to add further clarification that the reviewer thinks may be necessary.

---

> ### Author Response · Authors · 2025-11-23
> **Rebuttal to Reviewer 7WMM**
>
> > **Do H-Nets scale better because of Mamba?**
>
> In general, we note that there are two independent phenomena that operate independently:
> - **Mamba layers improve chunked/tokenized models:** Prior work has shown that interleaving Mamba/SSM layers with Transformer layers in a hybrid model outperforms isotropic Transformer-based tokenized models [A, B, C]. We find that the same holds for H-Net, where making the main network (i.e. the part of the model that operates on chunked inputs) hybrid improves H-Net performance as well (Fig. 13).
> - **Dynamic chunking is an improvement over standard tokenization:** Due to the above observation, we conducted our main experiment (Fig. 2) only by comparing between models that processed fully chunked inputs with pure Transformer-based networks.
>
> We do use Mamba layers in the encoder/decoder modules, as we believe they assist with the compression aspects of chunking, but this is distinct from the gain of hybrid BPE-based models over Transformer-based BPE models.
>
> [A] Tri Dao and Albert Gu. “Transformers are SSMs: Generalized models and efficient algorithms through structured state space duality”. In ICML (2024).\
> [B] Qwen Team. “Qwen3-Next: Towards Ultimate Training & Inference Efficiency”. https://qwen.ai/blog?id=4074cca80393150c248e508aa62983f9cb7d27cd&from=research.latest-advancements-list (2025).\
> [C] Kimi Team. “Kimi Linear: An Expressive, Efficient Attention Architecture”. In arXiv:2510.26692 (2025).
>
> > **“Multiplying upsampled vectors by their confidence scores incentivizes the routing module to make confident, accurate decisions.” However, they are multiplied by one, and the difference is only in the backward. Why does this trick help?**
>
> The main role of multiplying the confidence scores with the upsampled vectors is to provide additional training signals for the router (see Switch Transformer [D], which has a similar mechanism). However, we noticed that directly multiplying by confidence scores incentivized the model to increase the magnitude of the main network outputs, which could lead to instability.
>
> [D] Fedus et al. “Switch transformers: Scaling to trillion parameter models with simple and efficient sparsity”. In JMLR (2022).

---

### Official Review · Reviewer_jxd8 · 2025-11-01

**Soundness:** 3
**Presentation:** 3
**Contribution:** 3
**Rating:** 6
**Confidence:** 4

**Summary:**

The paper introduces a new tokeniser-free architecture, H-Net, that learns segmentation strategies and can be applied more than once, thereby creating learnable tokenisation. It evaluates its performance with respect to other architectures when data- and compute-matched.

**Strengths:**

- Interesting architecture that contributes to investigating the long-lasting issue of tokenisation in language models (aka tokenisation-free architectures)
- Many carefully conducted experiments, with positive results
- Paper clear and well written

**Weaknesses:**

- I think it is not ideal that the state of the art is only briefly described in the main part of the paper, and discussed at more length in an appendix. It makes the main part of the paper not really self-contained
- Moreover, the discussion on what is different and novel with H-Net with respect to previously published works is insufficient, especially in the main part of the paper. The authors write that "H-Nets […] unlock the ability to remove another layer of pre-processing, such as tokenizers, and instead learn them end-to-end". To me, it is not the case that H-Nets "unlock" (i.e. make possible for the first time ever) such an "ability". Previous work have done this before, and the authors should acknowledge it better, and clearly explain how and why H-Nets are novel and better.
- The authors rightly explain that such tokeniser-free architectures remove the need for heuristic tokenisation strategies, and create "optimal" (in some sense) segments that can be visualised and analysed. The authors only show a handful of examples in English, although they mention the fact that improvements are better in Chinese. This is certainly due to the fact that their algorithm re-discovers whitespace-separated tokens in English, but cannot to that in Chinese, where there are no whitespaces. But a whole discussion on the boundaries endogenously learned by the model is missing. In languages using a script that includes whitespaces, how often are boundaries placed on whitespaces? When are whitespaces not used as boundaries, and when does the model places boundaries at other places? In languages using a script that does not includes whitespaces, do the endogenously learned segments "look like" what linguistic traditions (and more recently, treebank developers) have defined as "words" or "word-forms"? When using 2 levels of hierarchy, does the first level generate morph-like units (it seems that it does not, at least on the examples shown on English)? To me, adding such a discussion is absolutely necessary to understand what the model actually learns and how (and when) it (should) perform(s) better than models relying on heuristic and/or statistics-based tokenisation strategies (whitespace, BPE, etc.)

**Questions:**

- will the codebase be released?

---

> ### Author Response · Authors · 2025-11-23
> **Rebuttal to Reviewer jxd8**
>
> We sincerely thank the reviewer for their constructive evaluation of our work. We appreciate the recognition of our contribution to advancing tokenization-free architectures, the acknowledgement of our carefully conducted experiments with strong results, and the comments on the clarity and quality of the writing.
>
> We now address the reviewer’s key concerns:
>
> > **The state of the art is only briefly described in the main part of the paper, and discussed at more length in an appendix. It makes the main part of the paper not really self-contained.**
>
> We appreciate the feedback! We would like to note that our extended related work (App. B) is 4 pages long and could not possibly fit in the main body. We are happy to add more discussion to the main body of any particular methods that the reviewer thinks should be expanded on, especially given a potential extra page.
>
> > **"H-Nets […] unlock the ability to remove another layer of pre-processing, such as tokenizers, and instead learn them end-to-end". To me, it is not the case that H-Nets "unlock" (i.e. make possible for the first time ever) such an “ability”**
>
> Our goal for H-Net was to satisfy the following desiderata, which is our definition of a language model that operates without tokenization **end-to-end**:
> - learning boundaries end-to-end (jointly with the entire model in one phase of training, just as standard LMs do)
> - able to support multi-level hierarchies (a natural consequence of being truly end-to-end)
> - matching or better scaling to isotropic baselines
>
> We believe that the paper’s introduction conveyed these goals, and to the best of our knowledge, no previous work simultaneously satisfies all of these desiderata. For example, prominent previous tokenizer-free approaches such as SpaceByte and BLT achieve strong performance but do not satisfy our definition of *end-to-end*. On the other hand, there exist methods that are technically end-to-end but have poor performance, making it debatable whether they have actually overcome the need for tokenization.
>
> That said, we are happy to iterate on the wording if the reviewer has specific suggestions.
>
> > **More explanations about boundaries.**
>
> We agree with the reviewer that adding more discussion of boundaries, on top of Sec. E.3.3., will help readers better understand the behavior of H-Nets. To our understanding, the reviewer is interested in the qualitative behavior of the model with respect to whitespaces, in particular:
>
> - In languages **without whitespaces**, does it discover semantically meaningful units?
> - In languages with **whitespaces**, is it learning more interesting units?
>
> To address these, we perform new qualitative visualizations in two different settings:
> - **Chinese text (w/o whitespaces).** H-Net reliably groups bytes into characters, and further composes single characters into multi-character words.
> - **English text (w/ whitespaces).** While H-Net does utilizes whitespace, it exhibits two desirable behaviors:
>     - For long words or technical terms, H-Net discovers segmentations that are more semantically coherent than BPE.
>     - H-Net frequently merges multiple consecutive words that form a meaningful phrase or semantic unit.
>
> To demonstrate how exactly H-Net draws boundaries for Chinese, we used a paragraph from the Chinese Wikipedia page about Mamba, and visualized the boundaries on a 1-stage H-Net at trained at Large scale: [https://postimg.cc/MfWH9pGW]. The model primarily segments individual characters into their own chunks, but is also able to find a few multi-character words such as 模型 (model), 创新 (innovation), and 计算 (compute), among others.
>
> As we have already presented examples of English boundaries in the paper, we further investigate how H-Net places boundaries on OOD inputs. For English examples, we used part of a paragraph from a recent paper on *de novo gene design* [A] selected for containing a lot of OOD gene names, and one the LaTeX source for the abstract of a recent arXiv *math preprint* [B]. Using the XL-size 2-stage H-Net, the chunks in these visualizations are the large chunks fed into the main network (i.e. after two levels of compression), and we obtained the following boundaries:
>
> - https://postimg.cc/JsM8Q78v [A]
> - https://postimg.cc/5XgVKc97 [B]
>
> These boundaries display some improved semantic segmentations (e.g. “bioinformatics”) compared to the BPE tokenizers, and a few multi-word merges (“a high”, “level of”, “the positive”, “the construction”, “the square”). Additionally, on technical terms such as gene names or “AlphaEvolve”, H-Net learns smoother segmentations than the BPE tokenizers.
>
> [A] Merchant et al. “Semantic design of functional de novo genes from a genomic language model”. In Nature (2025).\
> [B] Tao. “New Nikodym set constructions over finite fields”. In arXiv:2511.07721 (2025).
>
> > **Will the code be publicly released?**
>
> Yes, we will publicly release the code, which also can be found in the supplementary material.

---

### Official Review · Reviewer_675s · 2025-11-01

**Soundness:** 3
**Presentation:** 4
**Contribution:** 3
**Rating:** 6
**Confidence:** 4

**Summary:**

This paper introduces H-Net, a novel and compelling hierarchical architecture for end-to-end sequence modeling directly from raw bytes, aiming to replace the conventional tokenization pipeline. The core contribution is a "Dynamic Chunking" (DC) mechanism that learns content- and context-dependent segmentation strategies jointly with the main model. This is achieved through a clever combination of a similarity-based routing module to predict boundaries and a smoothing module to ensure stable, differentiable end-to-end training. The authors demonstrate empirically that H-Net not only trains stably at scale (up to 1.3B parameters) but also outperforms strong, compute-matched BPE-tokenized Transformer baselines. Furthermore, they show that the architecture can be recursively stacked (2-stage H-Net), leading to even better performance and scaling, particularly on languages and modalities where traditional tokenization heuristics are less effective, such as Chinese, code, and DNA sequences.

**Strengths:**

- **Novelty and Significance:** The paper's primary strength lies in presenting a robust and scalable framework for fully end-to-end, learnable segmentation. The Dynamic Chunking mechanism, especially the smoothing module that turns a discrete selection problem into a differentiable one, is an elegant solution to a notoriously difficult problem that has hampered previous efforts. This work represents a significant step towards realizing the "bitter lesson" by replacing a major handcrafted heuristic (tokenization) with a learned component.

- **Empirical Rigor:** The experimental evaluation is thorough and convincing. The authors conduct carefully controlled comparisons against strong baselines (BPE Transformer, MambaByte, SpaceByte) by matching both data and computational (FLOPs) budgets. The consistent outperformance of the 2-stage H-Net across different model scales is a powerful result.

- **Generality and Robustness:** The paper convincingly demonstrates the model's advantages beyond standard English text. The superior performance on Chinese, code, and DNA sequences validates the claim that a learned chunking strategy is more generalizable than fixed heuristics. The improved robustness to textual perturbations is another key benefit of operating directly on bytes.

- **Strong Ablation Studies:** The authors provide detailed ablation studies that validate their key architectural choices. These studies effectively demonstrate the importance of the smoothing module, the similarity-based routing, and the use of SSMs (Mamba-2) in the outer encoder/decoder layers, strengthening the credibility of the proposed design.

**Weaknesses:**

- **Practical Efficiency Concerns:** The paper candidly acknowledges that the current implementation can be up to 2x slower during training and has dynamic memory usage, which can be unpredictable. This is a significant practical hurdle for widespread adoption and large-scale training, as it complicates hardware optimization and resource allocation.

- **Uncertainty and Potential Fragility at Extreme Scale:** While stability is demonstrated up to 1.3B parameters, the fact that larger scales are left as future work raises questions about the fundamental robustness of the mechanism. The complex interplay between the main prediction loss and the auxiliary ratio loss could introduce unforeseen instabilities at much larger scales (e.g., 70B+). This limitation implies that the current mechanism, while effective, might not yet be a "fundamental" solution but rather one that is proven to work within a specific regime.

- **Need for Stronger Evidence on Principled Operation:** This is the most critical weakness. The paper's core claim is that it introduces a principled, end-to-end chunking mechanism. However, the primary evidence comes from ablation studies showing that removing a component (e.g., the smoothing module) degrades the final performance metric (BPB). This demonstrates that the components are necessary for good performance, but it does not sufficiently prove that they are working as theorized. For instance, it is unclear whether the performance gain is due to the smoothing module correctly interpolating uncertain boundaries, or if it's acting as a complex, yet effective, form of regularization. The visualizations of learned boundaries are a good first step, but more direct, quantitative evidence is needed to establish that this is a truly fundamental innovation rather than a highly effective, scale-specific heuristic.

- **Hyperparameter Sensitivity:** The ratio loss weight, $\alpha$, is fixed at 0.03 for all experiments without an accompanying ablation study.1 The model's performance and learned compression ratio could be highly sensitive to this value, potentially requiring extensive tuning for new domains or modalities. This replaces one form of tuning (tokenizer design) with another, potentially more opaque one.

**Questions:**

To further strengthen my assessment and address the concerns about the mechanism's fundamental nature, I would appreciate the authors' response to the following questions:

- **Evidence for Principled Operation:** Beyond the final performance metrics in ablation studies, can you provide more direct evidence that the core modules are functioning as hypothesized? For example:
    - **Routing Module:** Can you analyze the relationship between the cosine similarity scores and human-annotated semantic/syntactic boundaries? Does the "w/o cosine routing" variant learn a different, perhaps less interpretable, chunking strategy, or does it fail to learn any consistent strategy at all?
    - **Smoothing Module:** Does the smoothing module primarily act on low-confidence boundaries ($P_t \approx 0.5$), as intended? Could you provide statistics on the distribution of $P_t$ values and show how the EMA application correlates with them? This would help differentiate its role from being a general regularizer.

- **Scaling and Stability:** Could you elaborate on the potential stability challenges at scales significantly larger than 1.3B parameters? Have you observed any trends in the training dynamics (e.g., the interplay between the ratio loss and the main loss) that might suggest future issues, and do you have any hypotheses on how to mitigate them? Proving the mechanism's fundamental nature requires confidence in its scalability.

- **Hyperparameter $\alpha$:** Could you provide more intuition on the choice of $\alpha=0.03$? How sensitive is the model's final performance and, more importantly, the stability of the learned compression ratio to this hyperparameter? An ablation, even a small one, would be very valuable to understand the robustness of the training process.

**Details Of Ethics Concerns:**

No ethic review needed.

---

> ### Author Response · Authors · 2025-11-23
> **Rebuttal to Reviewer 675s**
>
> We sincerely thank the reviewer for their thorough and insightful evaluation of our work. We greatly appreciate the positive feedback on the novelty of our fully end-to-end learnable segmentation framework, the rigor of our empirical comparisons, the generality and robustness across diverse modalities, and the strength of our ablation studies.
>
> We now address the reviewer’s key concerns:
>
> > **Practical efficiency and scaling to larger settings**
>
> Training H-Net at larger scales (> 3B) is a very exciting future direction. Although we mentioned up to 2x slowdown in the paper, it was an extremely conservative estimation: at the XL (1.3B) scale, **we find that 1-stage H-Net is around 1.18x slower than the BPE transformer baseline, and 2-stage H-Net is around 1.37x slower. Our model’s data efficiency is more than these factors, so it still provides a significant wall clock-time improvement**. We are working on variants which we believe can operate with minimal slowdowns.
>
> Regarding scaling behavior, we found it very encouraging that H-Net 1.3B outperforms the baseline transformer earlier than in the corresponding 760M experiment; we are generally hopeful that H-Net’s advantage over BPE transformers improves with scale. We also note that, even for the standard transformer recipe, improvements for stable training at scale are still being developed (see e.g. qk-norm [A, B], MuonClip [C], attention sinks [D]). Thus, while we do expect some open problems to training very large-scale H-Nets, we also believe that the improvement gains will be well worth it.
>
> [A] Henry et al., “Query-Key Normalization for Transformers”. In EMNLP (2020).\
> [B] Yang et al., “Qwen-3 Technical Report”. In arXiv:2505.09388 (2025).\
> [C] Kimi Team, “Kimi K2: Open Agentic Intelligence”. In arXiv:2507.20534 (2025).\
> [D] Xiao et al., “Efficient Streaming Language Models with Attention Sinks”. In ICLR (2024).
>
> > **Behavioral difference between w/ and w/o cosine routing**
>
> To examine whether the default router with cosine similarity helps detecting semantically meaningful boundaries, we trained two 1-stage H-Nets on a Chinese dataset (similar to to App. D.4), one using a cossim-based routing and one without (router setup analogous to Fig. 7).
>
> We used a paragraph from the Mamba (architecture) Chinese Wikipedia page to visualize boundaries, producing the following boundaries: https://postimg.cc/WF5X4tCH. The router with cosine-similarity based routing identifies primarily character boundaries (with a few multi-character words), while the non-cosine router produces more segmentations that cut a single character into multiple pieces.
>
> > **Could you provide statistics on the distribution of  values and show how the EMA application correlates with them?**
>
> We are also interested in understanding the mechanistic advantage of the EMA. In our ablation study (Fig. 7), we show that not only is the EMA crucial for performance, it also helps stabilize the downsampling ratios significantly. Mechanistically, the EMA module’s smooth interpolation allows for better learning of confidence levels (App. C.2.1., Fig. 4).
>
> > **Hyperparameter (alpha) sensitivity.**
>
> We used alpha=0.03 for both the Large and XL scales, as we generally believed the scaling laws did not require extreme tuning of the alpha parameter. Running with alpha=0.01, 0.03, and 0.05 at the 2-stage Large scale yields the following:
>
> | alpha | L1/L0 | L2/L1 | BPB |
> | ---- | ----- | ----- | -------- |
> | 0.01 | 0.386 | 0.444 | 0.736 |
> | 0.03 | 0.355 | 0.384 | 0.743 |
> | 0.05 | 0.347 | 0.359 | 0.749 |
>
>
> These H-Nets with different alphas all train stably, and as one would expect, lower alpha leads to better performance at the cost of worse compression (and thus higher compute usage). In general, from a loss-vs-FLOPs scaling perspective, we do not believe that extremely precise tuning of alpha is required for the method.

---

### Official Review · Reviewer_UxoT · 2025-11-03

**Soundness:** 3
**Presentation:** 4
**Contribution:** 4
**Rating:** 8
**Confidence:** 3

**Summary:**

The authors propose a tokenizer-free architecture that dynamically segments sequences by jointly learning context and content dependent boundaries alongside the language modeling objective. Their model is hierarchical, enabling it to capture multiple levels of abstraction, from finegrained details to higher-order structure. In experiments on English, the architecture demonstrates increased robustness at the character level compared to BPE-based tokenizers, and the authors further report improvements for Chinese, code, and DNA sequences.

**Strengths:**

- The paper addresses an important limitation of many tokenizer-free architectures: training instability when boundary predictors must make discrete decisions (with or without supervision). Their proposed architecture is elegant in how it handles segmentation via the novel routing and smoothing mechanism.
- The paper is well written with lots of ablations, detailed discussions on different architectural and experimental choices that potentially aid reproducibility.
- Their experimental results are great to see, they demonstrate improvements over traditional BPE in downstream tasks, with robustness on character-level tasks, code etc. This speaks directly to the benefits of their dynamic tokenization strategies.

**Weaknesses:**

- Of course, this paper is not framed as a multilingual one, but the authors do claim improvements in other languages, and only evaluate on Chinese. While the improvements are notable, many recent frontier LMs are trained on web data across several languages. Do you have insights on how your architecture scales in a multilingual setting, when it is very common to have very distinct tokens mixed in individual sequences?
- In addition to what I mentioned above, I am particularly curious about the potential challenges of scaling HNET experiments to a truly multilingual setting involving languages with distinct scripts, varying data sizes, and diverse linguistic structures. When data is highly imbalanced across languages or domains, how robust is the quality of the learned segmentation? Even if prior ratios are fixed to guide boundary prediction convergence, could the quality of learned segmentation degrade as language-specific data becomes more limited, given that boundary learning is inherently context-dependent? I will be interested to hear your thoughts on this.
- Why are there no comparisons to BLT (https://arxiv.org/pdf/2412.09871) and the dynamic token pooling paper (https://arxiv.org/abs/2211.09761)? At the very least, there is some similarity in architectural designs with the dynamic token pooling paper.
-  There’s also no discussion about how well the model handles out-of-domain sequences, and I don’t mean big shifts going from natural language to DNA or code, but more subtle, realistic shifts like moving to scientific text or just long-tail words in the pretraining data. What kind of segmentations are observed? Are they optimal?

**Questions:**

- Any thoughts on how your model performs at really small scales <700M . I know that there seems to be more value these days in training models at larger scales, just curious if you have any thoughts.

---

> ### Author Response · Authors · 2025-11-23
> **Rebuttal to Reviewer UxoT**
>
> We sincerely thank the reviewer for the thorough and thoughtful evaluation of our work. We greatly appreciate the recognition of the novel routing and smoothing mechanisms in H-Nets for stabilizing discrete boundary decisions, as well as the positive feedback on our extensive ablations, detailed architectural discussions supporting reproducibility, and the strong empirical gains over BPE baselines across diverse tasks.
>
> We now address the reviewer’s key questions:
>
> > **How would H-Nets behave in multilingual settings?**
>
> We thank the reviewer for raising this insightful question. To examine the multilingual performance of H-Nets compared to tokenizer-based models, we used the same 220k-step protocol as in the main experiments to train both a Llama3-tokenizer based transformer and 2-stage H-Net at the XL scale.
>
> We obtained the data scaling curves shown here: https://postimg.cc/qtBg4RTP.
>
> We hypothesize that learning good boundaries/chunks is harder in the multilingual setting. Consequently, H-Net requires more data to learn serviceable boundaries. At the same time, a harder chunking problem also leads to a greater advantage for more expressive chunking methods that can handle the necessary complexity. Indeed, we observe that in the multilingual setting H-Net requires more data to become competitive with a tokenizer-based transformer model, but ultimately **achieves even more data efficiency than in the English case**.
>
> > **Why are there no comparisons to BLT and DPT?**
>
> BLT [A] and dynamic pooling transformer (DPT, [B]) are indeed important tokenizer-free approaches, and we therefore discuss them in Sec. B.1.3 (originally moved to the appendix due to page limits).
>
> On the empirical side, we did attempt to include them but found systematic comparison difficult:
>
> - DPT becomes unstable when scaled to the larger model sizes we consider (760M and 1.3B parameters), preventing us from obtaining reliable results at these scales.
> - BLT requires a multistage training process, including training a separate smaller model for entropy estimation, which makes it difficult to implement and tune.
>
> Instead, we primarily compared to SpaceByte and variants, which historically has presented as a strong baseline. Specifically, DPT evaluated entropy, space, and end-to-end variants, and reported that the whitespace variant was competitive with their end-to-end method ([B], Tab. 2, Fig. 4). Similarly, BLT’s space-based variant performs very closely to their entropy-based method ([A], Fig. 6).
>
> From this, we conclude that the space heuristic is very strong for English, and empirically competitive with these other methods (DPT and BLT). Our method manages to outperform space-based variants, in part due to our ability to achieve true multistage hierarchical modeling.
>
> [A] Pagnoni et al., “Byte Latent Transformer: Patches Scale Better Than Tokens”. In ACL (2025).\
> [B] Nawrot et al., “Efficient Transformers with Dynamic Token Pooling”. In ACL (2023).

---

> ### Author Response · Authors · 2025-11-23
> **Rebuttal to Reviewer UxoT**
>
> > **How does H-Net behave out of distribution? If training data is imbalanced, or input sequence is out-of-domain, could the quality of boundaries degrade?**
>
> We thank the reviewer for this interesting question. One hypothesis would be for H-Nets to produce more boundaries on OOD words, because an unexpected word is “harder”/requires more compute to process. However, more easily compressed examples (e.g. “aa…a” w/ length 1000) should lead to fewer boundaries and greater compression. Thus the exact boundary-drawing behavior is data-dependent and it is challenging to make any explicit claims.
>
> To perform a more empirical study, we looked at the XL-size 2-stage H-Net boundaries on two examples. (Here, the chunks are the large chunks fed into the main network after two levels of compression.) For our examples, we used part of a paragraph from a recent paper on *de novo gene design* [C] selected for containing a lot of OOD gene names, and one the LaTeX source for the abstract of a recent arXiv *math preprint* [D]. We obtained the following boundaries:
> - https://postimg.cc/JsM8Q78v [C]
> - https://postimg.cc/5XgVKc97 [D]
>
> These boundaries display some improved semantic segmentations (e.g. “bioinformatics”) compared to the BPE tokenizers, and a few multi-word merges (“a high”, “level of”, “the positive”, “the construction”, “the square”). In the first example, we also find that the first instance of a novel word (“AcrIIA2”) is split into 3 chunks (“Acr”+”IIA”+”2”), the second instance is 2 chunks (“AcrIIA” + “2”), and the last instance is one chunk. The same phenomenon appears with “Nikodym” in the second example. This pattern suggests that as the model “learns” a new word, it can become more capable of compressing it. One might consider this a form of in-context token learning, though a more thorough investigation is needed before making any strong conclusions.
>
> [C] Merchant et al. “Semantic design of functional de novo genes from a genomic language model”. In Nature (2025).\
> [D] Tao. “New Nikodym set constructions over finite fields”. In arXiv:2511.07721 (2025).
>
> > **How would H-Nets perform at smaller scales such as <700M?**
>
> Although H-Nets’ effectiveness becomes more noticeable when configuring the models with enough capacities, H-Nets still outperforms the baselines at the reasonably small scale: FLOPs matched to GPT-3 Medium (350M). With the same experimental setting used for Figure 3 (matching compute and data), the results are as shown below:
>
> | Model | Tokenizer | Architecture | d_model (D) | F-Edu BPB |
> | -------- | -------- | -------- | -------- | -------- |
> | Transformer | GPT2 | T24 | 1024 | 0.794 |
> | H-Net (1-stage) | 9-DC | M4+T26+M4 | 768, 1024 | 0.792 |
> | H-Net (2-stage) | (3,3)-DC | M4+T1M4+T24+M4T1+M4 | 768, 768, 1024 | 0.787 |

---

### Comment · Area_Chair_ng7L · 2025-11-23
**Review-Author discussion**

Hi Reviewers,

Please kinly and actively participate in the review-author dicussion, raise your further concerns so that the authors can explain more, and make your final decisions.

---

### Author Response · Authors · 2025-11-23
**General Response**

We thank all reviewers for their helpful feedback and constructive suggestions, as well as for recognizing both the novelty of our method and the strength of our experimental results.

Several reviewers raised a common concern that the main paper is not sufficiently self-contained. In response, we have prepared an improved, revised version of the manuscript, in which all new additions to the main text are highlighted in red.

Multiple reviewers also asked questions about boundaries, with a view towards better understanding the chunking behavior of H-Net. Prompted by such questions, we evaluated and visualized the chunking boundaries of our 2-stage XL H-Net on a few examples:

- Scientific (biology) text: https://postimg.cc/JsM8Q78v
- Math/LaTeX code: https://postimg.cc/5XgVKc97

These examples show principled chunking decisions, ability to form multi-word chunks, and the capability to adapt chunking decisions over time. We hope that these examples will help enhance reviewers’ understanding of H-Net’s behavior. We have elaborated further on these results in response to more specific questions and generally thank reviewers for raising this point.

---

### Meta-Review · Area_Chair_7jzR · 2026-01-08

**Summary:**

The paper proposes H-Net, an end-to-end hierarchical sequence model that learns dynamic chunk boundaries for language models, and shows that when compute/data are matched it can beat strong BPE Transformer baselines, with multi-stage hierarchy improving scaling and robustness across domains (e.g., Chinese/code/DNA). The work is genuinely different from predictable incremental papers and is backed by strong controlled results agreed by all reviewers.

**Reviewer Concerns:**

Addressed: Multilingual experiment, OOD, small model

Outstanding: memory

**Reviewer Scores:**

The reviewers who gave 6 might further raise the score.

---

### Decision · Program_Chairs · 2026-01-26

Accept (Poster)